# In hot water: Uncertainties in projecting marine heatwaves impacts on seagrass meadows

**Paula S. Hatum**[ORCID][1]*, **Kathryn McMahon**[2], **Kerrie Mengersen**[ORCID][1], **Jennifer K. McWhorter**[3], **Paul P.-Y. Wu**[1]

**1** School of Mathematical Sciences, Centre for Data Science, University of Technology, Brisbane, Queensland, Australia, **2** School of Science and Centre for Marine Ecosystems Research, Edith Cowan University, Joondalup, Western Australia, Australia, **3** NOAA Atlantic Oceanographic and Meteorological Laboratory, Miami, Florida, United States of America

\* paula.sobenkohatum@hdr.qut.edu.au

**Data Availability Statement:** All relevant data are within the manuscript and its Supporting information files.

## Abstract

Seagrass ecosystems, vital as primary producer habitats for maintaining high biodiversity and delivering numerous ecosystem services, face increasing threats from climate change, particularly marine heatwaves. This study introduces a pioneering methodology that integrates Dynamic Bayesian Networks of ecosystem resilience with climate projections, aiming to enhance our understanding of seagrass responses to extreme climate events. We developed cutting-edge metrics for measuring shoot density and biomass in terms of population and site extinction, presented as annual ratios relative to their respective baselines. These metrics include associated uncertainties and projected recovery times. This innovative approach was applied in a case study focusing on *Zostera muelleri* in Gladstone Harbour, Australia. Utilising five downscaled climate models with a 10 km resolution, our study encompasses a range of Shared Socioeconomic Pathways and emissions trajectories, offering a comprehensive perspective on potential future scenarios. Our findings reveal significant variations in seagrass resilience and recovery times across different climate scenarios, accompanied by varying degrees of uncertainty. For instance, under the optimistic SSP1-1.9 scenario, seagrass demonstrated a capacity for recovery heat stress, with shoot density ratios improving from 0.2 (90% Prediction Interval 0.219, 0.221) in 2041 to 0.5 (90% PI 0.198, 1.076) by 2044. However, this scenario also highlighted potential site extinction risks, with recovery gaps spanning 12 to 18 years. In contrast, the more pessimistic SSP5-8.5 scenario revealed a significant decline in seagrass health, with shoot density ratios decreasing from 0.42 (90% PI 0.226, 0.455) in 2041 to just 0.2 (90% PI 0.211, 0.221) in 2048, and no recovery observed after 2038. This study, through its novel integration of climate models, Dynamic Bayesian Networks, and Monte Carlo methods, offers a groundbreaking approach to ecological forecasting, significantly enhancing seagrass resilience assessment and supporting climate adaptation strategies under changing climatic conditions. This methodology holds great potential for application across various sites and future climate scenarios, offering a versatile tool for integrating Dynamic Bayesian Networks ecosystem models.

**Funding:** Payment for the submission of this manuscript was supported by the Centre for Data Science, Queensland University of Technology (QUT).

**Competing interests:** The authors have declared that no competing interests exist.

## 1 Introduction

Extreme climatic events have gained significant attention due to their notable effects on ecosystem structure and dynamics [1]. Marine heatwaves, a coherent area of warm sea surface temperature (SST) that persists for days to months [2], have become more frequent and intense over the past century [3–5], impacting the integrity of marine ecosystems globally [6]. Marine heatwaves can cause widespread mortality [7, 8], dramatic range shifts of marine species, and community reconfiguration [9–11], and unexpected adverse effects on ecosystem services associated industries, such as commercial fisheries [12]. In this context, projections of the impacts of climate change on marine ecosystems are essential for the development of effective adaptation strategies, but these projections come with uncertainty, complicating long-term planning and risk assessment [13].

Seagrasses, marine flowering plants that provide vital ecosystem services and functions in nearshore coastal environments (e.g., food through fisheries, control of erosion, and protection against floods) [14–16], are experiencing a worryingly global decline. Although the primary cause of this decline has been largely attributed to the degradation of water quality in many areas (e.g., dredging, sediment loading, and eutrophication), increasing evidence indicates that Extreme climatic events, such as marine heatwaves, have also played a significant role [17–20]. Some recently observed marine heatwaves revealed the high vulnerability of seagrass and other coastal foundation species to such extreme climate events [21–23].

Although the resistance and recovery of seagrasses to heat stress induced by marine heatwaves is not well understood, research has shown that these processes can vary across species and possibly among populations of the same species within regions as a function of their thermal tolerances [24]. Generally, meadows formed by slow-growing and long-living species typical of temperate climates, such as *Posidonia*, *Amphibolis*, and *Zostera*, are highly susceptible to climate change disturbances, while smaller colonising species common of tropical regions, such as *Halodule*, *Halophila*, and *Syringodium*, are more resistant to warming and marine heatwaves [25].

Given the ecological importance of seagrass in maintaining high biodiversity and a range of other ecosystem services, and with Extreme climatic events predicted to become more frequent and intense [26], understanding their impacts on the marine ecosystem is critical for assessing species' adaptive capacity under future climate change scenarios. However, predicting natural ecosystem responses to stressors can be particularly challenging due to their often highly variable and complex nature and uncertainty surrounding their dynamic responses [27–30]. These stressors rarely occur in isolation and generally cause impacts through the combinations of multiple abiotic and biotic drivers. For example, elevated temperatures resulting from global warming and heatwaves [22] can affect coastal ecosystems, which are frequently compounded or intensified when combined with other stress factors [31, 32]. Cumulative impacts can play an essential role in determining the current and future state of seagrass ecosystems and may not be predicted from the individual effect of each variable operating in isolation [32–35].

Climate impact studies on marine ecosystems have increased in recent years due to the availability of climate model simulations and various approaches for incorporating climate forcing in predictive models. Earth System Models, General Circulation Models, and Regional Climate Models connected to fishing and Marine Ecosystem Models are employed to assess the impact of climate variability and change on Living Marine Resources [36–39]. These models attempt to capture ecosystem dynamics via species distribution models, ecological models structured based on species interactions and energy transfer across trophic levels or using hybrid models. While existing approaches to predict species and ecological responses to climate change offer valuable insights [40–43], they are not without limitations.

Three key gaps exist in current marine ecosystem modeling approaches [40–43]. First, existing models often focus on a limited set of processes affecting ecosystem dynamics, thereby neglecting other potentially influential factors. Second, these models are generally not equipped to capture or predict the non-linear and dynamic responses of entire marine ecosystems to various perturbations. This is a significant limitation given that ecosystems are inherently complex, characterized by non-linear dynamics, threshold effects, and limited predictability. Third, the challenge of incorporating uncertainty from future climate data into these predictive models remains largely unresolved. These gaps in the modeling structure underscore the need for a more comprehensive approach that can account for the dynamic nature of ecosystems, capture non-linear responses to environmental changes, and effectively integrate uncertainties from climate projections.

In addition to the complexity of ecosystem responses to climate change described above, there are also several sources of uncertainty in climate projections that can hinder the reliability and accuracy of these models. These uncertainties arise from various sources in climate projections, including model structural differences, initial conditions, scenarios, parameters, and resolution/bias correction [44], and can limit the robustness and precision of the model projections [45, 46]. When these projections are used to evaluate environmental systems, this uncertainty can exacerbate ecological model uncertainty, which stems from the variability in model type, design, and parameterization [47]. Knowledge of how to incorporate climate uncertainty into ecological analyses remains challenging in climate change research [13, 45, 48–51].

Various techniques have been developed and applied to quantify different sources of uncertainty associated with climate change. These techniques include multi-model ensembles, model emulation, model sensitivity analysis, expert elicitation for the evaluation of structural uncertainty, as well as Monte Carlo simulations for input data uncertainty evaluation [52, 53]. Ensemble modeling has emerged as a valuable method for capturing climate change impacts and communicating uncertainties more effectively [54, 55]. This approach involves developing a set of models, with each model member representing different working hypotheses or alternative formulations of uncertain processes.

Directly linking climate models with ecosystem models facilitates the development of management actions, enhances our understanding of potential causal pathways, and allows for an explicit characterization of uncertainty. However, the process of coupling these models can be particularly challenging due to their inherent complexities and the additional uncertainties that arise when they are interconnected [45]. Although a fully integrated socio-ecological climate model can capture interactions across the complete systems-of-systems, such complexity (i.e., large parameter space) makes robust calibration and evaluation of the model difficult [56, 57]. There is a need for more comprehensive treatments of uncertainty [58], including relationships/feedback between climate and its effects when linking climate and impacts. A significant opportunity for innovation lies at the intersection of integrating climate models with ecosystem models in a probabilistic framework. One promising approach is through the use of Bayesian Networks coupled with climate models.

Bayesian Network (BN) has been applied as a tool for capturing different sources of uncertainty in complex systems [49, 59–61]. BNs have been employed extensively across various environmental sectors such as Integrated Water Resource Management, ecology, conservation, maritime spatial planning, fisheries, and agronomy [62–73]. However, the application of BNs for modeling ecological and environmental systems in the context of climate change remains limited. A select number of studies [74–80] have utilized BNs to investigate the impact of climate change on essential natural resources. These studies have covered various issues, from water scarcity and soil erosion to biodiversity loss, eutrophication, and sea-level changes. BNs

have proven effective in exploring model uncertainty, assessing the impacts of management interventions on climate scenarios, and even in predicting future conditions of coral reefs, as demonstrated in studies by [49, 59–61].

However, a limitation of BNs is their inherent static characteristic, which restricts their applicability in modeling temporal dynamics [59]. To address this, a coupling approach combining System Dynamics and BNs has been developed to simulate the system over time [60]. While SD aids in understanding the non-linear behavior of complex systems, it is less adept at handling uncertainty compared to BNs, which are capable of handling both quantitative and qualitative data but may fall short of capturing feedback within a dynamic system. Furthermore, although Bayesian Network structures, direct acyclic graphs, can facilitate the systematic identification of variables within multidisciplinary subsystems, it can still be challenging to incorporate all the factors into the modeling procedure.

Dynamic Bayesian Networks (DBNs) provide one approach for integrating factors under uncertainty for systems that require a dynamic representation over both time and space [79, 81]. DBNs are an extension of standard Bayesian Networks that use the Object-Oriented Programming paradigm to divide the time period of interest into discrete stages and replicate the network structure for each step [82]. These networks are then linked in such a way that information can be transferred from one-time slice to the next one, allowing for a dynamic understanding of the system over time. One challenge with DBNs is the relative complexity and specialized statistical skills required, which explains their scarcity in the current literature [79].

The primary aim of this paper is to develop a methodological framework that integrates ecosystem DBNs with climate model projections to assess the resilience of seagrass ecosystems. By doing so, we provide an explicit quantification of uncertainty as part of an end-to-end model combining climate scenarios with ecosystem resilience responses to support management. These tools contribute to enhanced risk modeling and interpretation of uncertainty for climate impacts [83, 84]. The approach employed here accommodates both quantitative data, results of empirical analyses, and qualitative data obtained from expert opinions. By utilizing the inherent strengths of DBNs in the explicit modeling of factor relationships between factors using conditional probability, we obtained the predicted probability of resilience outcomes and the factors contributing to it, including the climate scenario. Such a method is particularly advantageous for assessing cumulative effects involving multiple risk factors, as it provides explicit modeling of uncertainty, which is essential in long-time-horizon predictions associated with climate change.

## 2 Materials and methods

### 2.1 Gladstone Harbour case study

Our case study includes a *Zostera muelleri* seagrass meadow located at Pelican Banks, Gladstone Harbour (latitude: 23˚45′58″S and longitude: 151˚18′30″E), Australia (Fig 1). As in the broader Great Barrier Reef World Heritage Area, the rainfall distribution in Gladstone is highly seasonal [85], with a summer wet season (December–April) followed by an extended dry season (May-November) [86]. At this site, sea surface temperature (SST) varies between a minimum of 19.2˚C (August) and a maximum of 29.7˚C (January), averaging 24.4˚C across the year [87]. Gladstone Harbour is subtropical and reflects a transition zone between temperate and tropical species. Pelican Banks represents the largest extent of seagrasses in Gladstone Harbour, where the tropical subspecies *Zostera muelleri* forms a predominantly monospecific intertidal seagrass meadow on mud banks.

Seagrass evolution in the Port of Gladstone is influenced by a multitude of environmental and anthropogenic factors, including the local tidal and wave regimes. The area experiences

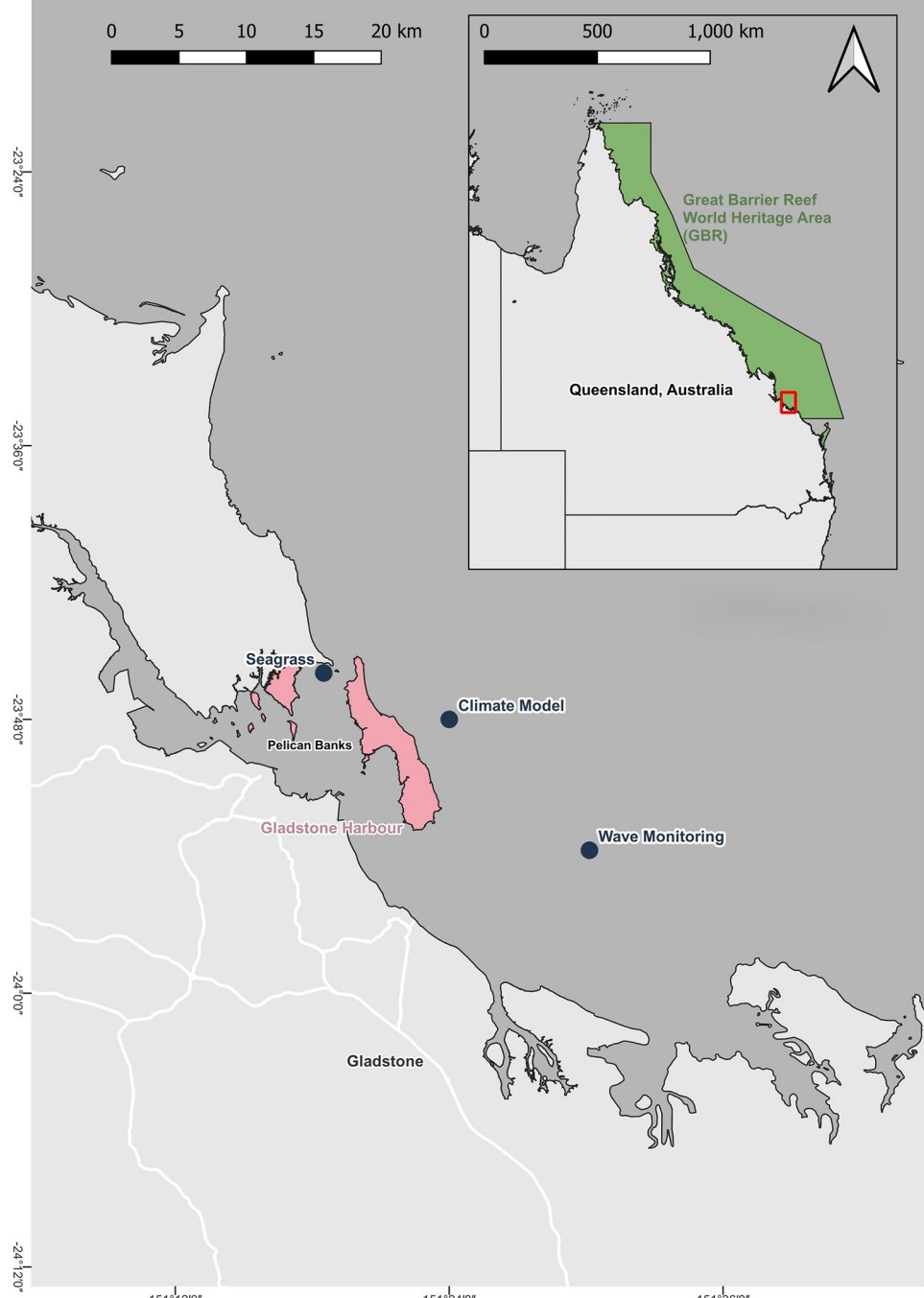

**Fig 1. Map highlighting the study area in Gladstone, Pelican Bank, Australia.** The locations are marked as follows: 1) Seagrass—Indicate *Z. muelleri* meadow location; 2) Wave Monitoring—Designates sites where sea surface temperature data was collected at high sample rates through wave monitoring; and Climate Model: Marks where sea surface temperature projections were made using an ensemble of climate models.

variations in wave energy, with more energetic waves typically occurring during high tides, which play significant roles in the dispersal patterns of seagrass propagules [88, 89]. Frequent natural disturbances such as tropical storms, grazing by dugongs and turtles, and bioturbation enhance local clonal diversity by creating gaps that seeds colonize, preventing monopolization

by competitively superior genotypes [90, 91]. Additionally, anthropogenic disturbances like dredging, land reclamation, and nutrient enrichment often increase sexual reproduction and genotypic diversity in seagrass meadows [92, 93]. High genotypic diversity is crucial for resilience to disturbances [94]. Genetic surveys reveal significant structuring among sites, influenced by the area's hydrodynamic complexity, resulting in varying connectivity and isolation [95, 96].

Projections indicate that sea-surface temperatures in the Great Barrier Reef could rise by 1˚C to 3˚C by 2100, with summer water temperatures potentially exceeding 33˚C, well above the current distributional range of *Z. muelleri* [97]. Seagrass populations in Gladstone have shown varying tolerance to heat stress. For instance, *Z. muelleri* from Gladstone exhibited high sensitivity to 33˚C, suffering almost complete mortality after four weeks, whereas populations from Midge Point showed greater heat tolerance, maintaining positive productivity at 35˚C and increasing thermal optima to 37.3˚C [98]. Despite declines, such as those in southeast Queensland, *Z. muelleri* meadows often recover within three years [99, 100].

## 2.2 Seagrass Dynamic Bayesian Network model

In this study, the seagrass DBN model is an extension of the general DBN framework for seagrass resilience (S1 Fig), as described by [101]. The original model, developed using data and expert knowledge of 25 global sites across various seagrass genera, primarily evaluates seagrass resilience to light and burial stressors associated with dredging. Recognizing the original model's limitation in addressing heat stress, our study has extended the model to include the effects of heat stress scenarios, especially those resulting from marine heatwaves. Moderate heat stress can cause carbon imbalance in seagrasses, as their respiration increases proportionally more than their photosynthesis. This can result in irreversible damage to their photosynthetic apparatus if the stress exceeds certain thresholds, as noted by various studies [102–104].

Building on the methodology introduced by Hatum et al. [105], our model captures the impact of marine heatwaves on seagrass resilience through two principal mechanisms. The first mechanism focuses on the optimal temperature range essential for plant growth, detailing the adverse effects on physiological status when temperatures deviate from this range. Such deviations can impair normal plant functions and disrupt the seed recruitment process, with younger plant stages being particularly vulnerable to elevated temperatures [106]. The second mechanism, heat stress, accounts for mortality or a reduction in shoot density resulting from temperatures that exceed the tolerance limits of the seagrasses. To model these effects, a temperature node is linked to physiological status and seed recruitment, while a heat stress node quantifies the loss in biomass, reflecting the direct impact of severe heat on seagrass mortality (Fig 2).

The original model, designed for *Zostera muelleri* at Pelican Banks, Gladstone Harbour, was developed with input from a group of experts and has had its structure and parameters validated as reported in Wu *et al.* [107]. The model was updated and validated to incorporate the effects of temperature and heat stress [105] based on the guidelines in Hatum *et al.* [108]. The spatial-temporal resolution for this model is monthly timesteps, which correspond to the growth dynamics of seagrass for inference purposes, and it is spatially calibrated for a single seagrass meadow. The seagrass DBN model captures the conditional probabilistic relationships among population variables (e.g., biomass, shoot density), factors relating to resistance (e.g., growth and physiology), recovery (e.g., physiology, seed, and vegetative growth), site conditions (e.g., genera present and location characteristics such as depth, climate, and tidal regime) and environmental factors (e.g., light and water quality) (Fig 2).

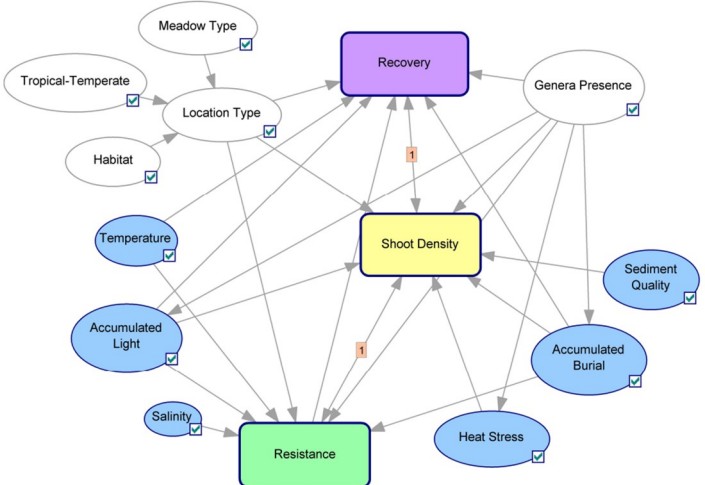

**Fig 2. Dynamic Bayesian Network representation.** Ovals represent nodes, and arrows indicate causal parent-child relationships, with the absence of a link suggesting conditional independence. Subnetworks are shown in rounded rectangles [77]. This representation is adapted from [101] and developed using GeNIe.

## 2.3 Integrating climate model projections with the DBN model

**2.3.1 Measurement and projections of temperature data.** Our first step involves the selection and preparation of the primary input parameters for the DBN model: temperature and heat stress. These parameters serve as the two key mechanisms for capturing the impact of marine heatwave on seagrass ecosystems. Daily sea surface temperature projections up to the year 2100, considering four Shared Socioeconomic Pathways (SSPs) as described below, were used in this study. The use of SSPs introduced a range of potential future climate conditions, thereby adding a layer to assess one component of uncertainty, the uncertainty of future temperatures based on climate models.

The Coupled Model Intercomparison Project Phase 6 (CMIP6) provides climate predictions by integrating a combination of land-use scenarios and emissions. These predictions are based on SSPs and revised versions of the Representative Concentration Pathways. These SSPs, outlined in the latest IPCC assessment report, consider future socioeconomic changes and climate mitigation strategies, thus expanding on the Representative Concentration Pathways concepts. For accurate local-scale projections, particularly in the shallow coastal waters inhabited by seagrass, Jennifer K. McWhorter utilized the semi-dynamic downscaling technique from Halloran *et al.* [109] to refine five CMIP6 models (MRI-ESM2-0 [110], EC-Earth3-Veg [111], UKESM1-0-LL [112], CNRM-ESM2-1 [113], IPSL-ESM2-0 [114]) to a 10 km resolution across four SSPs. This 1-D, semi-dynamic downscaling method inputs atmospheric variables (winds, downwelling shortwave and longwave radiation, pressure, humidity, air temperature) while accounting for local bathymetry and tidal conditions to perform a vertical water column calculation outputting SSTs [109, 115, 116]. The projections employ an ensemble mean, calculated as the average of all climate projections, with each projection being referred to as an ensemble member. This was necessary as General Circulation Models typically have a 1° horizontal resolution, which lacks tidal mixing or near-shore dynamics. A specific grid cell was selected for the seagrass meadow under study, located at 23°48′S and 151° 24′E.

The five scenarios used in our study are also in the IPCC's latest report [83] (SSP1.9, SSP1-2.6, SSP2-4.5, SSP3-7.0, SSP5-8.5), predicting temperature rises of 1.5°C by 2040 and 3.3-5.7°C

by 2100 under the highest emission scenario. Our study focuses on three SSP trajectories (SSP1, SSP3, SSP5) and four emissions paths (SSP1-1.9, SSP1-2.6, SSP3-7.0, SSP5-8.5) [117], with the latter numbers indicating peak radiative forcing ($W/m^2$). SSP1-1.9, within SSP1, targets limiting warming to 1.5˚C using advanced $CO_2$ removal technologies [118]. Notably, this scenario overshoots the 1.5˚C target, approaching nearly 2˚C before returning to 1.5˚C by the end of the century. SSP3 emphasizes regional rivalry and self-sufficiency, leading to slower growth and increased inequalities. In contrast, SSP5 envisions a fossil-fuel-driven world with robust economic and population growth driven by competitive markets. The CMIP6 datasets used to inform these scenarios were accessed through the Centre for Environmental Data Analysis. Additionally, downscaling details are available via the Zenodo repository, as outlined in Halloran *et al.* [109]. The CMIP6 data can be accessed at https://catalogue.ceda.ac.uk, and the downscaling information is available at Zenodo DOI: 10.5281/zenodo.4147559.

The downscaled CMIP6 climate models generate daily SST projections based on a range of SSPs, offering appropriate estimates for the temperatures experienced by intertidal seagrasses. Unlike their subtidal counterparts that remain continuously submerged, these intertidal species are periodically exposed to air, but the models estimate their overall temperature exposure using SSTs, not air temperature, even though they are calculated from air temperature [109]. Note that 'daily average' temperatures refer to the average computed for each individual climate model projection rather than an ensemble mean across all models. This methodology tends to smooth out peak daily temperatures that might otherwise indicate potential heat stress. Therefore, to supplement the CMIP6 models' daily SST projections, we incorporated high-frequency SST data from a wave-monitoring buoy near Gladstone, accessible via the Coastal Data System—Waves website http://www.qld.gov.au/waves. These measurements, taken every 30 minutes, covered 2014-2016 and 2021-2022, providing a detailed view of daily temperature variations crucial for understanding the thermal stress on intertidal seagrasses. Data from 2017 to 2019 were excluded due to formatting issues. High-frequency temperature data facilitated the establishment of thresholds for temperature and heat stress nodes (Materials and Methods, Section 2.3.3), effectively capturing daily variations. DBN inference discretization meets technical needs and identifies key decision thresholds, providing discrete probabilities for risk assessment [107, 119]. This coarse method effectively captures system uncertainties and knowledge [120].

Given the unpredictable nature of future human behavior, these scenarios present a spectrum of plausible climate futures, reflecting the outcomes of diverse policy choices. SSP1 advocates for reduced emissions and a shift towards renewable energy, promoting a more sustainable future [121]. In contrast, SSP3, which has a higher radiative forcing, suggests a future marked by increased pollution and limited progress in health and education [122]. SSP5 envisions a heavy reliance on fossil fuels, resulting in significant increases in greenhouse gas emissions [123]. It is important that these SSPs scenarios are not predictive and, therefore, do not carry associated probabilities [124]. Their primary purpose is to provide decision-makers with a range of potential outcomes derived from various plausible choices, thereby facilitating informed decision-making. As an example of how these projections can be used, consider the variability observed in the shoot density ratios of *Z. muelleri* under different SSP scenarios. This variability serves as a vital indicator, guiding strategic resource allocation in times of uncertainty and underscoring the importance of prioritizing interventions in regions most likely to experience significant impacts.

**2.3.2 Establishing thresholds for temperature and heat stress.** Once temperature data was secured, thresholds for temperature and heat stress variables were established using expert elicitation and peer-reviewed literature. These thresholds convert continuous variables into categorical variables relevant to environmental impact on plant physiology for management,

such as optimal or sub-optimal temperatures. Given the complexity of the seagrass system and the limitations of the data and knowledge about it, such discretization also reflects the resolution of information available. Thermal optima, as published for specific genera/species, served as the basis for defining temperature states.

The temperature thresholds used in the analysis were particularly informed by research conducted on *Zostera muelleri* in locations such as Cockle Bay and Picnic Bay on Magnetic Island, and Moreton Bay near the northern Great Barrier Reef—areas geographically proximate to Gladstone. Recognizing the limitations stemming from relying on data exclusively from these areas, especially given the absence of direct data from Gladstone itself, the thresholds were refined and validated through expert opinions. Parameters specific to the location, such as tropical or temperate distribution, were also factored in, given that certain seagrass genera/species exhibit broad distributional ranges.

Since photosynthesis, like other biological processes, is temperature dependent, the photosynthetic rates of aquatic plants incrementally rise with temperature until a thermal optimum is reached [103, 104]. Beyond this optimal point, the photosynthetic rates begin to decline at higher temperatures [125]. According to Collier *et al.* [126], the thermal optimum for net plant productivity of *Z. muelleri* was identified to be 24°C. Consequently, the sub-optimal temperature state was identified to fall above this value.

The second mechanism involves heat stress, which captures seagrass mortality during marine heatwaves. This is achieved by assuming that the temperature thresholds for seagrass mortality have been exceeded for a specified epoch of time (day(s) or month(s)). In a study conducted by [127], a decrease in leaf growth rate from 27°C was reported over a 5-day period, resulting in plant mortality when exposed to a temperature of 33°C for 30 consecutive days. Our study obtained temperature data from a site different from the meadow in [127], specifically near Gladstone. Consequently, the temperature thresholds indicative of seagrass mortality, as delineated by [127], surpassed the levels recorded in the high-frequency data of this study, wherein temperatures rarely reached or exceeded 33°C. Therefore, a heat stress threshold of 30°C for anticipated shoot mortality has been established in this study.

**2.3.3 Modelling temperature and heat stress.** The climate projections for the designated seagrass meadow include daily average temperature data. However, the DBN model operates on a monthly timescale. Consequently, this necessitates converting mean temperature data to align with the optimal temperature (thermal optimality) and shoot mortality thresholds (thermal extremes). This section describes the methodology of this conversion process.

*i. Thermal optimality.* Let $T_d$ be a binary variable with states sub-optimal and optimal, which is used to classify the temperature for a day $d$. Utilizing high-frequency logger data collected every 30 minutes and applying a threshold of 24°C, we classify a day as having an optimal temperature if all recorded temperature measurements for that day are less than or equal to 24°C. Conversely, the day is classified as sub-optimal if this condition is not met. Then, we can model the daily temperature optimality given a projected mean daily temperature, denoted as $SST_d$, using the following approach:

$$T_d = \log\left(\frac{p}{1-p}\right) = \beta_0 + \beta_1 \cdot SST_d \qquad (1)$$

where $p$ is the probability of a day being sub-optimal. The coefficients are determined through fitting. The probability $p_t^T$ of sub-optimal temperature for month $t$ is estimated

using the following equation:

$$p_t^T = \left(\frac{1}{n_t}\right)\sum_{\forall d \in t}(T_d)$$ (2)

where $t$ denotes month of a give year, $n_t$ represents the number of days within a given month. During Monte Carlo simulations, samples for each month are generated with the probability $p_t^T$ that the temperature for that month will be classified as sub-optimal; if not, it is classified as optimal.

ii. *Thermal extremes.* Let $Y_d$ denote the maximum temperature for a day $d$, derived from 30-minute interval data collected by loggers. To estimate the relationship between $Y_d$ and average daily temperature $SST_d$, we employed a simple linear regression model:

$$Y_d = \beta_0 + \beta_1 \cdot SST_d + \beta_2 \cdot t$$ (3)

where $t$ denotes the month of a given year and is incorporated as a factor to account for seasonal variations in the maximum temperature $Y_d$. $SST_d$ is the average daily temperature, and $\beta_0, \beta_1$ and $\beta_2$ are the regression coefficients. A day is classified as experiencing heat stress effect $Z_d = 1$ if $Y_d \geq 30$; otherwise, $Z_d = 0$. Let us define:

$$Z_t = \sum_{\forall d \in t} Z_d$$ (4)

where $Z_t$ denotes the number of days $d$ exceeding maximum temperature $Y_d \geq 30$ in a month $t$. Given the extreme nature of marine heatwaves, we adopted a conservative assumption that more than 15 days of heat stress in a month constitutes an entire month of heat stress, leading to the following expression for the probability of heat stress:

$$p_t^Z = \begin{cases} 1 & \text{if } Z_t \geq 15 \\ \dfrac{Z_t}{n_t} & \text{if } Z_t < 15 \end{cases}$$ (5)

where $n_t$ is the total number of days in the month. As before, during the Monte-Carlo simulation, samples for each month are generated based on the probability $p_t^Z$ that a heat stress effect occurs that month; otherwise, no heat stress is assumed. Note that this assumption has limited influence on the outcomes of this specific one-month case study, rendering shorter intervals unnecessary for our current analysis. However, this approach could be modified to include shorter-duration marine heatwave assessments at other sites or to investigate future deteriorations in the physiological conditions of this site.

## 2.4 Modelling seagrass resilience to heat stress

Ecological resilience serves as a crucial measure for evaluating the impact of climate change on seagrass. Different seagrass species can exhibit diverse responses to global change stressors, although species-specific characteristics may also influence the reactions. Key factors include the characteristics of disturbances, environmental conditions, and the presence or absence of feedback that impacts seagrass health, population structure, and reproductive capacity [25].

To quantify resilience, [107] focused on three significant ecological interactions extracted from the literature. Firstly, resistance characterizes an ecosystem's ability to withstand frequent and intense disturbances [128, 129]. Secondly, recovery reflects the duration for an ecosystem

to recover following a disruption [130]. Lastly, persistence assesses the probability of complete degradation over time when compared to baseline conditions, with higher values indicating an elevated risk of extinction [131, 132]. In this study, we defined two principal measures of resilience: the shoot density ratio, denoted as $r$, and the recovery time, represented as $q$. The ratio $r$ serves a dual role, assessing resistance by comparing shoot density against immediate disturbances to baseline conditions and evaluating persistence through the long-term maintenance of shoot density states (high, moderate, low, zero) under continuous heat stress. Further details on these metrics are presented in the following section.

**2.4.1 Annual changes in shoot density ratios.** Shoot density ratios are defined within the DBN for each state—high, moderate, low, and zero shoot density—as follows:

$$r_S(t) = \frac{p_S(t)}{p_S^B(t)} \tag{6}$$

where $S$ represents the state (high, moderate, low or zero), $B$ denotes the baseline scenario, which is established as the year 2022, and $t$ is measured in months.

The baseline year for this study was set as 2022, based on the availability of environmental and biological ecosystem data. Significantly, 2022 marked the third consecutive year of good condition for Gladstone Harbour's seagrass meadows, following poor conditions from 2015 to 2018, making it an ideal reference for current and future assessments [133]. This choice recognizes the potential for significant environmental and ecological shifts over extended periods, underscoring the necessity to adjust the baseline in future studies to reflect these long-term changes accurately and from a practical management standpoint.

The annual ratio is derived from the monthly ratios as their average, providing a more comprehensive view of the overall trajectory rather than merely reflecting seasonal variations:

$$r_S(y) = \frac{1}{12} \sum_{t \in y} r_S(t) \tag{7}$$

where $y$ is the year. A high zero shoot density ratio ($r_{zero}(y)$) is indicative of site extinction when compared to baseline levels, whereas an elevated high shoot density ratio ($r_{high}(y)$) signifies a high population density.

**2.4.2 Post-disturbance recovery time.** Recovery time is calculated as the period, expressed in years, within which the ratio first falls below 90% of the baseline and then rises to reach or exceed 90% of the baseline. Let $q(y)$ be the recovery time for year $y$, where $y$ is the first year when the ratio $r_S(y)$ falls below 90% of the baseline. Then $y'$ is the earliest subsequent year when the ratio $r_S(y')$ returns to or exceeds 90% of the baseline. The recovery time $q(y)$ is calculated as $q(y) = y' - y$, provided that $y' > y$, $r_S(y) < 0.9$, and $r_S(y') \geq 0.9$ (S4 Fig). This analysis focuses primarily on the recovery times for states of high and moderate shoot density. This focus is informed by the rationale that assessing recovery to zero and low states holds limited analytical value, as these states predominantly signify a decline in the seagrass meadow.

The final uncertainty aspect in this model was addressed by conducting Monte Carlo simulations for each climate scenario (n = 100). These simulations generate diverse outcomes through random sampling from probability distributions, offering a comprehensive perspective on potential future scenarios and their risks. The simulation process produced a set of ratios $r_S^i(y)$ for each year, with $i$ ranging from 1 to 100, each representing individual samples. From these samples, we calculate the mean and prediction intervals (PIs) to understand the expected outcomes and their variability. To better comprehend and visualize the distribution of $r_S(y)$, we consider two types of PIs. The first is a 90% interval, which spans from the 95th to the 5th percentiles, and the second is a 50% interval, ranging from the 75th to the 25th

percentiles (S1–S16 Tables). Additionally, we acquire the raw samples $p^i_{xs}(t)$ for each node state $xs$ at every month $t$ for each sample $i$. This data is used for explanatory inference to help infer what the most likely cause was for different resilience outcomes.

# 3 Results

The validation of the DBN model, as conducted by Wu *et al.* [101], involved a comparing its predicted probabilities with those from external datasets not utilized during its training phase. These external datasets were derived from studies focused on the effects of light deprivation on seagrass, specifically simulating the impacts of dredging. Wu *et al.* [101] reported that the mean-squared error (MSE) between the model's predicted-state probabilities and the observed values was in the range of 0.01 to 0.05. Prior to the integration of heat stress nodes, the model results were calibrated to align with baseline conditions as established by Wu *et al.* [101]. Subsequent to this calibration, the output from the DBN was subjected to qualitative evaluation through expert judgment, as detailed in Kragt, M. E. [134]. Additionally, the simulations of the downscaling climate models were validated using sea surface temperature data from satellite and mooring observations. Halloran *et al.* [109] identified a slight temperature bias, which may be attributed to the model operating as a one-dimensional vertical simulation that does not account for horizontal processes.

Detailed in Materials and Methods, Section 2.3.3, we developed a framework adapting daily average temperatures from climate models to our ecosystem model, using simpler yet effective regression models. This includes a logistic regression with a McFadden's $R^2$ of 0.7997 and a linear regression with an $R^2$ of 0.9937. By integrating Monte Carlo simulations and innovative resilience metrics, our framework enhances transparency and explainability and provides a clear depiction of uncertainties. The resilience outcomes for shoot density ratios and recovery times are presented in Sections 3.1 and 3.2, respectively, focusing on the case study of the *Z. muelleri* seagrass meadow in Gladstone, Australia. To illustrate the resilience and associated uncertainty in shoot density predictions relative to the baseline, the shoot density ratio for each state has been plotted over time (Figs 3 and 4, S1 and S3 Figs).

## 3.1 Annual changes in shoot density ratios

Under the SSP1-1.9 scenario, the annual high shoot density ratios exhibit considerable variability across the evaluated years, experiencing both peaks and troughs. For instance, in the year 2063, the annual density ratio reaches a high of 1.0008, while it plummets to a low of 0.2125 in 2067. Despite fluctuations, a general rebound is evident, especially after the mid-2080s. Percentile spreads are narrow in specific years like 2033 and 2063 but widen notably in the 2040s, 2070s, and late 2080s to early 2090s, indicating increased uncertainty during these periods (Fig 3, S1 Table). This pattern is consistent with the SSP1-1.9 scenario, which overshoots to 2°C of global average warming before returning to 1.5°C by the end of the century. In the SSP1-2.6 scenario from 2030-2099, annual shoot density ratios fluctuate but generally recover, except in the mid-2040s and mid-2060s. By the 2090s, a decline was also evident. Regarding PIs, they sometimes align closely with the annual ratio, for example, at the end of the 2030s, suggesting low uncertainty. However, toward the end of the 2050s, wider percentile spreads indicate elevated uncertainty (Fig 3, S2 Table).

The annual high shoot density ratio under the SSP3-7.0 scenario exhibits fluctuations until the mid-2060s, after which a decreasing trend is observed. Despite a long-term downward trend, averages rebound more frequently until the 2050s—a trend not observed in subsequent years. Prediction intervals remain wide until the end of the 2060s, after which they narrow, signifying high uncertainty from the 2030s through the 2060s and a decrease toward the end of

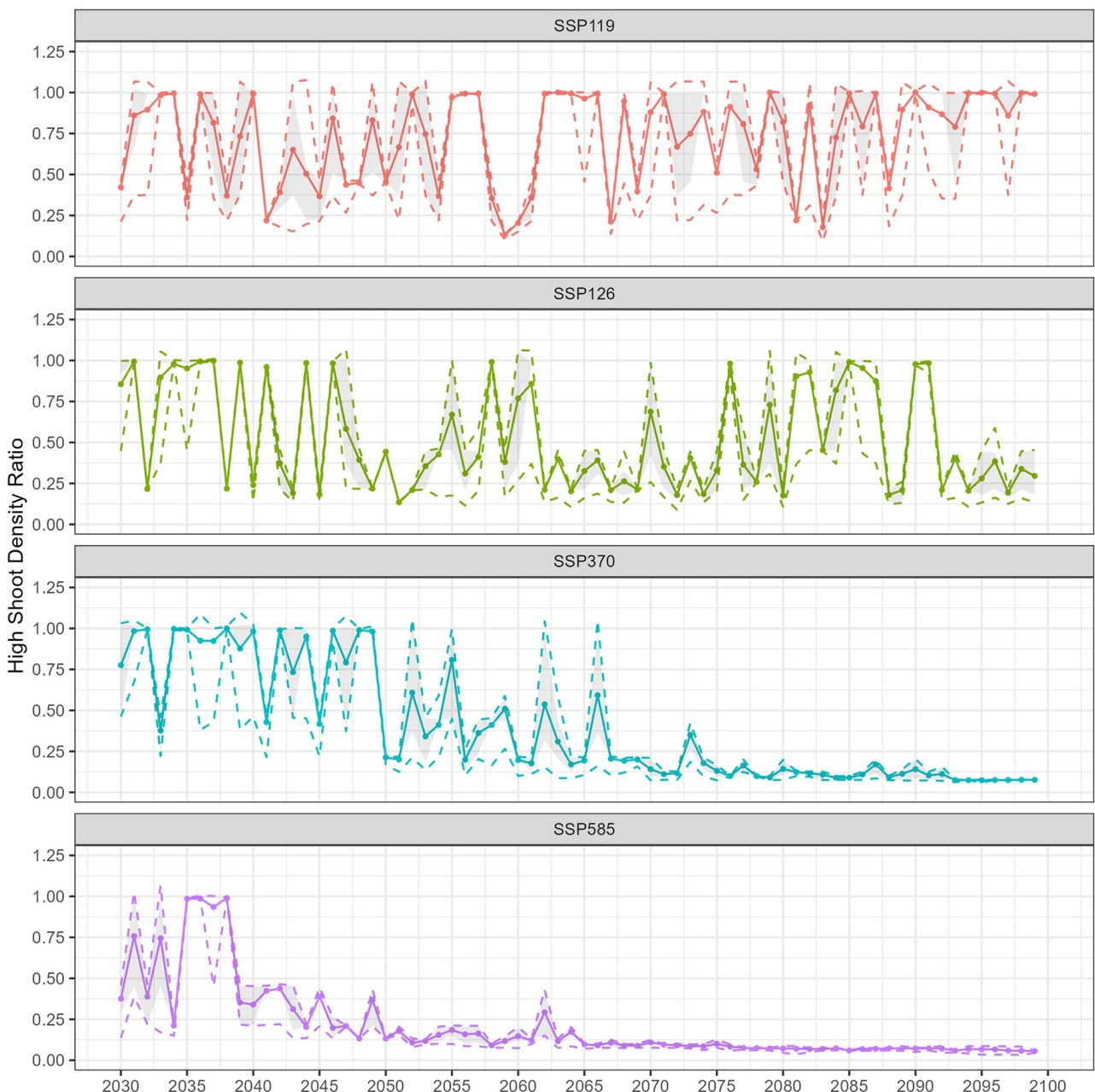

**Fig 3. Annual projections for high shoot density ratio of _Z. muelleri_ in Gladstone, Australia, depicting different socioeconomic pathway scenarios from 2030 to 2100.** Each colour corresponds to a distinct scenario: red (SSP1-1.9), green (SSP1-2.6), blue (SSP3-7.0), and purple (SSP5-8.5). The lines represent key statistical measures: solid lines for the average, dashed lines for the 95[th] percentile (upper) and 5[th] percentile (lower), a grey area for the 75[th] percentile (upper) and 25[th] percentile (lower) derived from a dataset of 100 samples.

the century (Fig 3, S3 Table). Similar to the SSP3-7.0 scenario, the SSP5-8.5 scenario shows a declining trend in annual high shoot density ratios. However, this decline began a decade earlier, in the early 2040s, compared to the 2050s in the SSP3-7.0 scenario. Initially, from 2030 to 2034, the ratios vary between 0.38 and 0.99. This is followed by relative stability in the mid-2030s. Post-2040s, the ratios generally decline to 0.25 or lower; although occasional minor

increases occur, they do not exceed 0.5. In the early 2030s, wider PIs indicate greater variability, but these intervals narrow significantly in later years (Fig 3, S4 Table).

Under the SSP1-1.9 scenario, annual zero shoot density ratios display variability over time without a consistent trend in average values. Although spikes occur, notably toward the end of the 2050s and in the mid-2060s, the meadow appears unlikely to be driven to zero under this scenario. As for the PIs, periods of high uncertainty generally coincide with peaks in the annual zero shoot density ratios (Fig 4, S13 Table). For the SSP1-2.6 scenario from 2030 to 2099, the annual zero shoot density ratios demonstrate substantial variability over the years, with no discernible upward or downward trend in average values. Specifically, the average values fluctuate significantly, ranging from a low of 0.8968 in the year 2050 to a peak of 3.0216 in 2051. Higher uncertainty is observed from the 2040s to the 2060s, followed by a narrower spread in the distribution of results from the 2060s to the 2090s. After that, uncertainty increases again (Fig 4, S14 Table).

In the SSP3-7.0 scenario, annual zero shoot density ratios generally remain around one from 2030 to 2050. These ratios undergo a significant increase starting from the 2050s, with values escalating and peaking in the 2090s. Alongside this upward trend, PIs widened beginning in the late 2050s. Although the spread is narrow both up to 2050 and in the late 2090s, it markedly widens from the 2050s onward. For example, the range between the $5^{th}$ to $95^{th}$ percentiles expands noticeably as we approach the end of the century (Fig 4, S15 Table). In the SSP5-8.5 scenario, average values generally rise post-2040, peaking in the late 2090s with values exceeding 7. This upward trend is similar to that in the SSP3-7.0 scenario, except it commences a decade earlier. From the 2040s onward, a widening spread is evident in both the PIs, becoming especially pronounced towards the end of the century (Fig 4, S16 Table).

Furthermore, our results indicate that even under the most adverse scenarios, the meadow population does not drop to zero. This finding is somewhat reassuring, suggesting that complete site extinction is not imminent. However, the increase in zero shoot density ratios, especially under the SSP3-7.0 and SSP5-8.5 scenarios, is a stark reminder of the potential risks. This trend suggests that, in the absence of shoots, local populations of the species in question could face extinction. The heightened uncertainty during these periods, as indicated by the wide ranges between the PIs, signifies the ecosystem's sensitivity to varying outcomes. For instance, the $r_{zero}(2050)$ for the SSP5-8.5 scenario is 3.0451, a substantial rise of approximately 221% from the 2049 average of 1.3958. This increase underscores a greater risk of site extinction. Moreover, the narrowing of the interval between the $5^{th}$ and $95^{th}$ percentile from 1.0132 to 2.2772 in 2049 to 3.0109 to 3.1168 in 2050, which indicates a decreased uncertainty in the zero shoot density ratio estimates, thereby providing a more precise indication of the site extinction risk during this period.

Heatmaps provide a visual representation of seagrass resistance, illustrating its capacity to endure increased severity and frequency of disturbance events (Fig 5). High and moderate shoot density ratios over time under different climatic scenarios are represented on heatmaps to assess resistance. Consistently maintaining high or moderate shoot density, for example, might indicate strong resistance. Fig 5a shows that in the SSP1-1.9 scenario, there is a potential to sustain higher population levels post-2060. The SSP1-2.6 scenario presents a more moderate trajectory, with a decline from the 2050s to the mid-2070s and another dip in the 2090s. Despite this decline in high population density, it is less pronounced compared to the SSP3-7.0 and SSP5-8.5 scenarios.

The SSP3-7.0 scenario maintains high shoot density ratio values up to the 2050s, after which a consistent decline ensues. Conversely, the SSP5-8.5 scenario begins a marked decline in 2039, continuing through the end of the century. The SSP1-1.9 and SSP1-2.6 scenarios display elevated moderate shoot density ratios, suggesting that even with a decline in high shoot

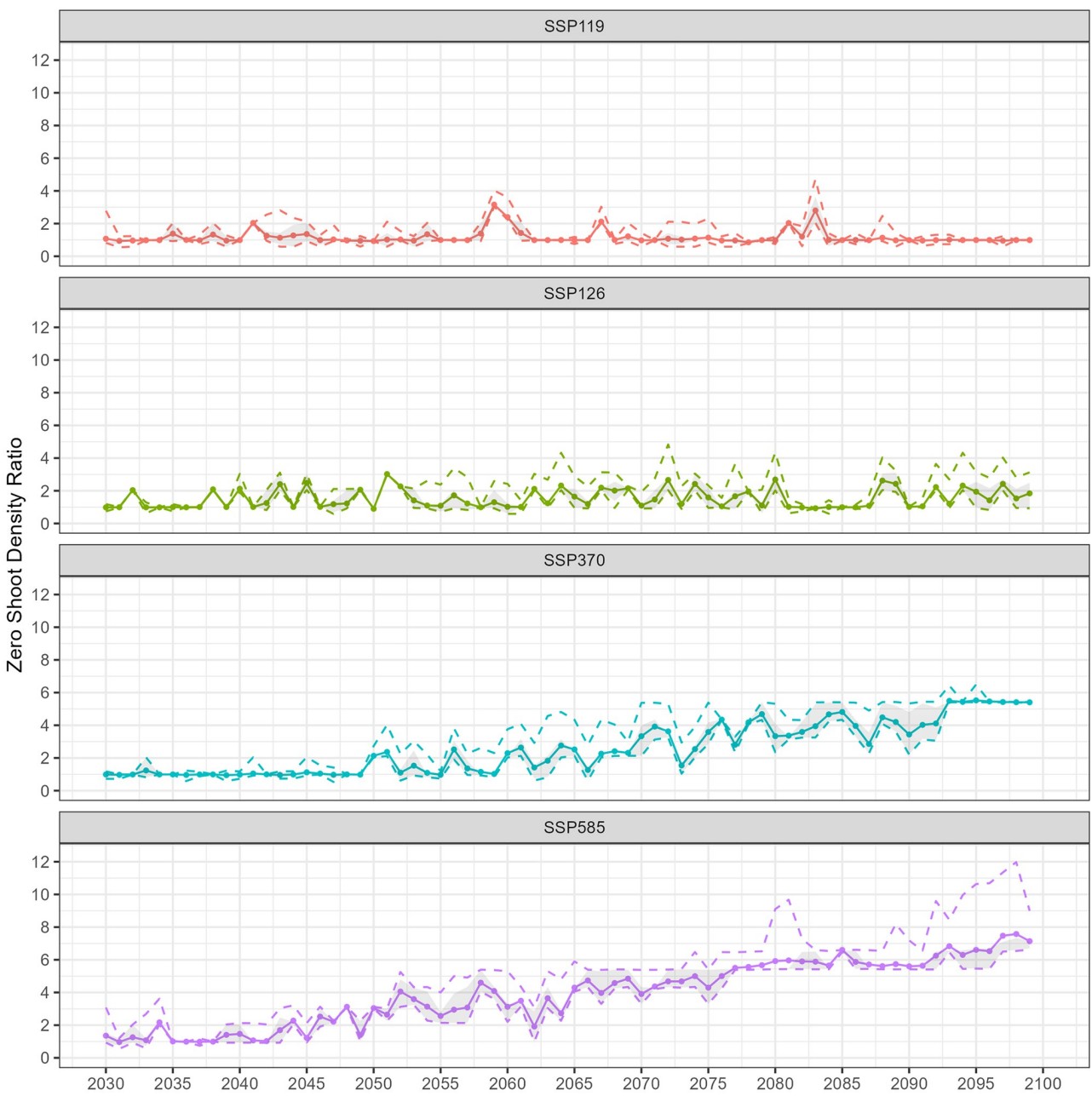

**Fig 4. Annual projections for zero shoot density ratio of *Z. muelleri* in Gladstone, Australia, depicting different socioeconomic pathway scenarios from 2030 to 2100.** Each colour corresponds to a distinct scenario: red (SSP1-1.9), green (SSP1-2.6), blue (SSP3-7.0), and purple (SSP5-8.5). The lines represent key statistical measures: solid lines for the average, dashed lines for the 95th percentile (upper) and 5th percentile (lower), a grey area for the 75th percentile (upper) and 25th percentile (lower) derived from a dataset of 100 samples.

density ratios, moderate levels are more sustainable under these scenarios (Fig 5b). In contrast, the SSP3-7.0 and SSP5-8.5 scenarios predict more challenging futures. The SSP5-8.5 scenario, in particular, indicates a rapid decline, pointing to a steeper trajectory with significant reductions in moderate shoot density ratios as the century advances.

Fig 5c and 5d showcase the low and zero shoot density ratios over time for various SSP. These heat maps provide a visual insight into the meadow's resilience and the potential site

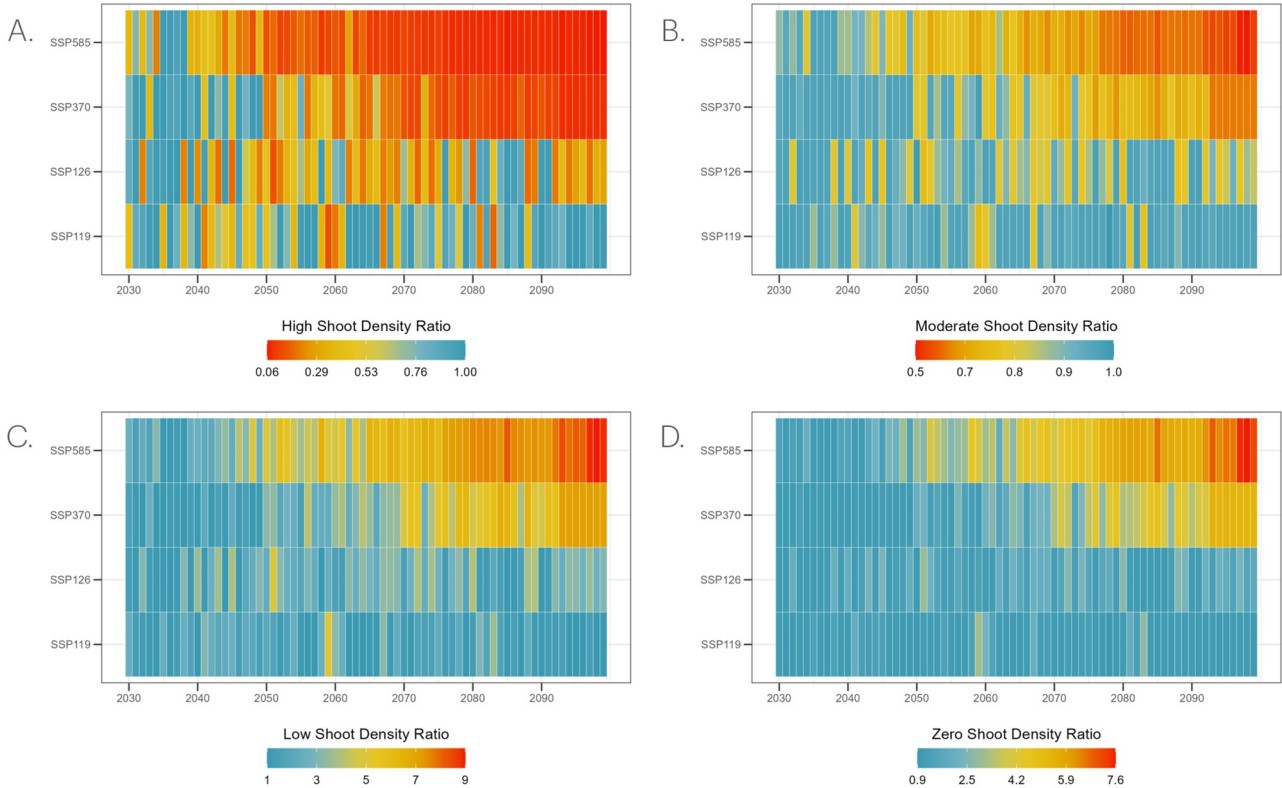

**Fig 5. Annual projections of *Z. muelleri* shoot density ratios in Gladstone, Australia, under various socioeconomic pathway scenarios from 2030 to 2100.** (a) and (b) represent High and Moderate shoot density ratios, where light blue indicates greater levels and dark red is lower than baseline. (c) and (d) represents Low and Zero shoot density ratios, where light blue indicates lower levels and dark red has greater levels than baseline.

extinction risk. In the SSP1-1.9 and SSP1-2.6 scenarios, the meadow demonstrates a reduced likelihood (represented in blue) of reaching zero or low population levels compared to the SSP3-7.0 and SSP5-8.5 scenarios. Notably, under SSP3-7.0, there is a consistent annual low and zero shoot density from the 2030s to the 2070s, followed by a gradual increase through 2099. The SSP5-8.5 scenario maintains a relatively stable ratio until the mid-2050s, after which it begins to rise.

## 3.2 Post-disturbance recovery time

In Fig 6 and Table 1, the recovery times for high shoot density ratios reveal distinct trends across SSPs. For SSP1-1.9, the average recovery time is 2.44 years, with the longest interval between recoveries being 12 years. This pathway experienced 27 recovery events in total, including four around the mid-2030s. After a 12-year gap, consistent recovery was observed from 2052 through the century's end. In the SSP1-2.6 pathway, two main recovery phases were identified: one from the 2030s to mid-2040s and another in the 2080s. The first phase was interrupted by a 12-year gap, concluding in 2058, and the second phase began 18 years later, in 2076, leading into the 2080s recovery period. This pathway averaged 3.53 years between recoveries, with a total of 16 events. The SSP3-7.0 pathway showed a faster recovery cycle, averaging 1.38 years between its 12 recorded events. In contrast, the SSP5-8.5 pathway had only four recovery events, all between 2036 and 2038, indicating a less frequent recovery pattern.

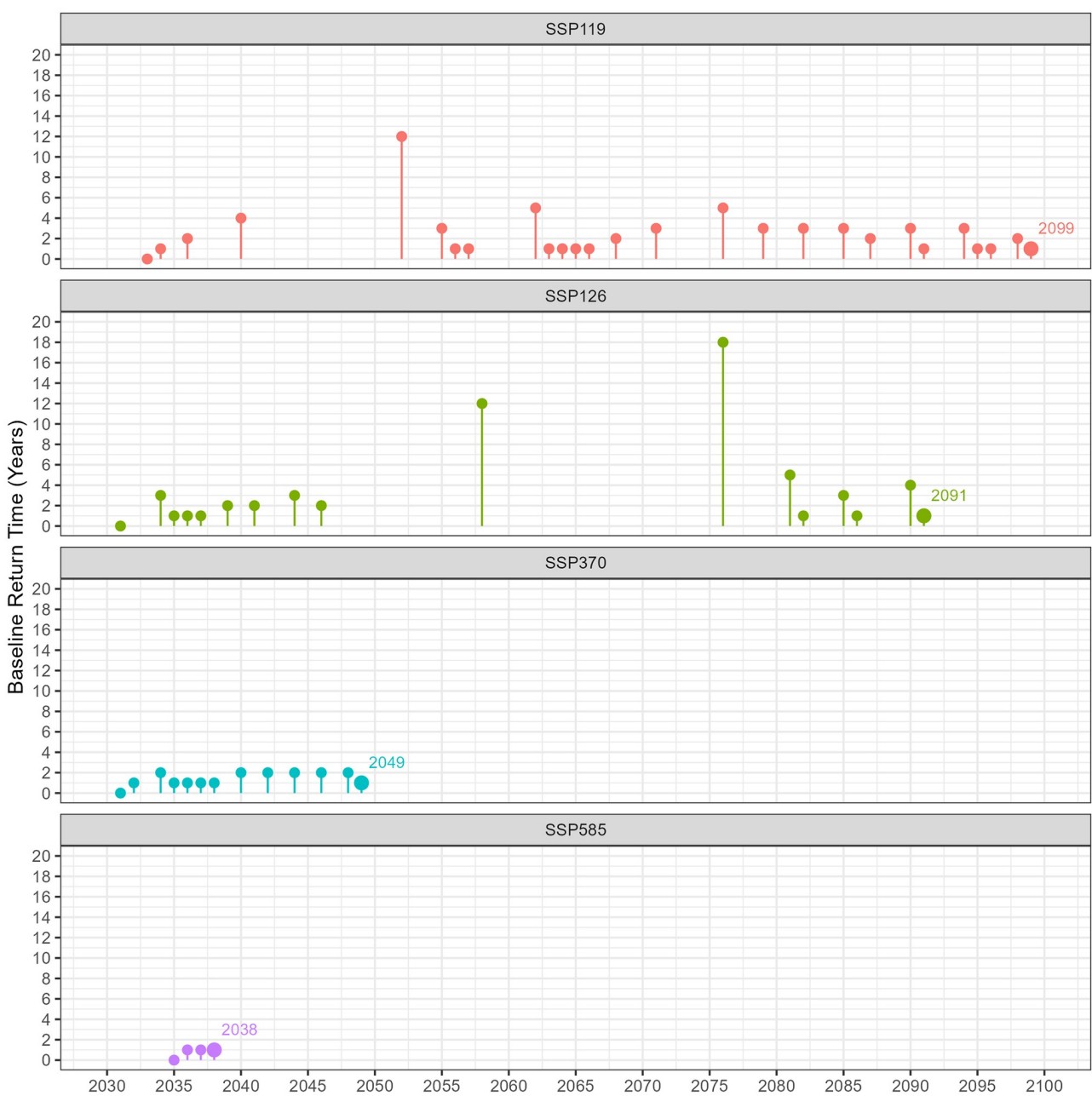

**Fig 6. Baseline return time for high shoot density ratios of *Z. muelleri* in Gladstone, Australia, depicting various socioeconomic pathway scenarios from 2030 to 2100.** Each color represents a distinct scenario, as follows: red (SSP1-1.9), green (SSP1-2.6), blue (SSP3-7.0), and purple (SSP5-8.5). The lollipop graphs illustrate the years since the last event where the high shoot density ratio was equal to or exceeded 0.9.

Significant variations are visible when focusing on moderate shoot density ratio recovery duration, as shown in Fig 7 and Table 1. The SSP1-1.9 pathway is remarkable for its 58 instances of returning to baseline conditions, which indicates a high frequency of recovery events across the entire period. A singular 5-year gap is observed in the early 2060s. Similarly, the SSP1-2.6 pathway exhibits an average recovery time of 1.69 years and is characterized by extended maximum recovery periods, notably from the 2060s to the 2080s and again in the

**Table 1. Recovery data for high and moderate shoot density ratios.**

|  | SSP1-1.9 | SSP1-2.6 | SSP3-7.0 | SSP5-8.5 |
|---|---|---|---|---|
| **High Shoot Density Ratio** | | | | |
| Mean Recovery Time (Years) | 2.44 | 3.53 | 1.38 | 1.38 |
| Max Gap in Recovery (Years) | 12 | 18 | 2 | 2 |
| Number of Recovery Instances | 27 | 16 | 12 | 4 |
| **Moderate Shoot Density Ratio** | | | | |
| Mean Recovery Time (Years) | 1.19 | 1.69 | 1.48 | 1.48 |
| Max Gap in Recovery (Years) | 5 | 4 | 7 | 7 |
| Number of Recovery Instances | 58 | 39 | 28 | 10 |

2090s, in contrast to the SSP1-1.9 scenario. For the SSP3-70 pathway, a total of 28 recovery events are observed. These events occur yearly from 2030 to 2049, after which they become more sparse. The final recovery event takes place in 2073, following a 7-year gap. The SSP5-8.5 pathway exhibits an average recovery time of 1.48 years per event and is characterized by fewer recovery instances, with 10 events recorded exclusively from 2030 to 2049.

When analyzing the recovery data for high versus moderate shoot density ratios, a notable trend emerges: the mean number of recovery events tends to be lower for moderate densities. Additionally, when comparing across the SSPs, optimistic scenarios (SSP1-1.9 and SSP1-2.6) show a higher average of recovery events occurring throughout the study period with some gaps. In contrast, pessimistic scenarios (SSP3-7.0 and SSP5-8.5) have fewer average recovery events, mostly concentrated early in the study period and more closely spaced. This difference in distribution and frequency of recovery events between optimistic and pessimistic scenarios highlights the varying impacts of these scenarios on the recovery patterns of seagrass meadows.

## 4 Discussion

This study introduces a novel methodological framework in response to the urgent need for effective risk and uncertainty communication in decision-making about complex ecosystems impacted by climate change. The framework is designed to enable long-term predictions of seagrass resilience by integrating climate model projections with a DBN. Through this integration, the framework generates a range of scenarios that illustrate different heatwave conditions within each SSP context of the case study. While the framework's application has been limited to a single location thus far, its effectiveness is evident. The framework not only demonstrates the practicality of the approach but also indicates that the innovative metrics it employs can be adapted for use in other regions and with different species, highlighting its broad applicability in ecological studies.

A notable distinction between the current study and the one conducted by [107] is the temporal scope for the model predictions and the resilience metrics derived from the model. [107] limited their work to a 12-year simulation window, whereas this study extended it to a 70-year projection. This extended timeframe consists of a 2-year model initialization phase, followed by a 70-year projection that considers variations in temperature and heat stress across four combinations of SSP scenarios (SSP1-1.9, SSP1-2.6, SSP3-7.0, and SSP5-8.5). To assess the resilience of seagrass within the context of long-term future climate projections, it was necessary to modify the existing resilience assessment framework. Given the seasonal dynamics of seagrass and the extended decadal timescale of climate change, the resilience assessment transitioned from a monthly to a yearly temp-step size. Similar to [101], resilience is assessed with

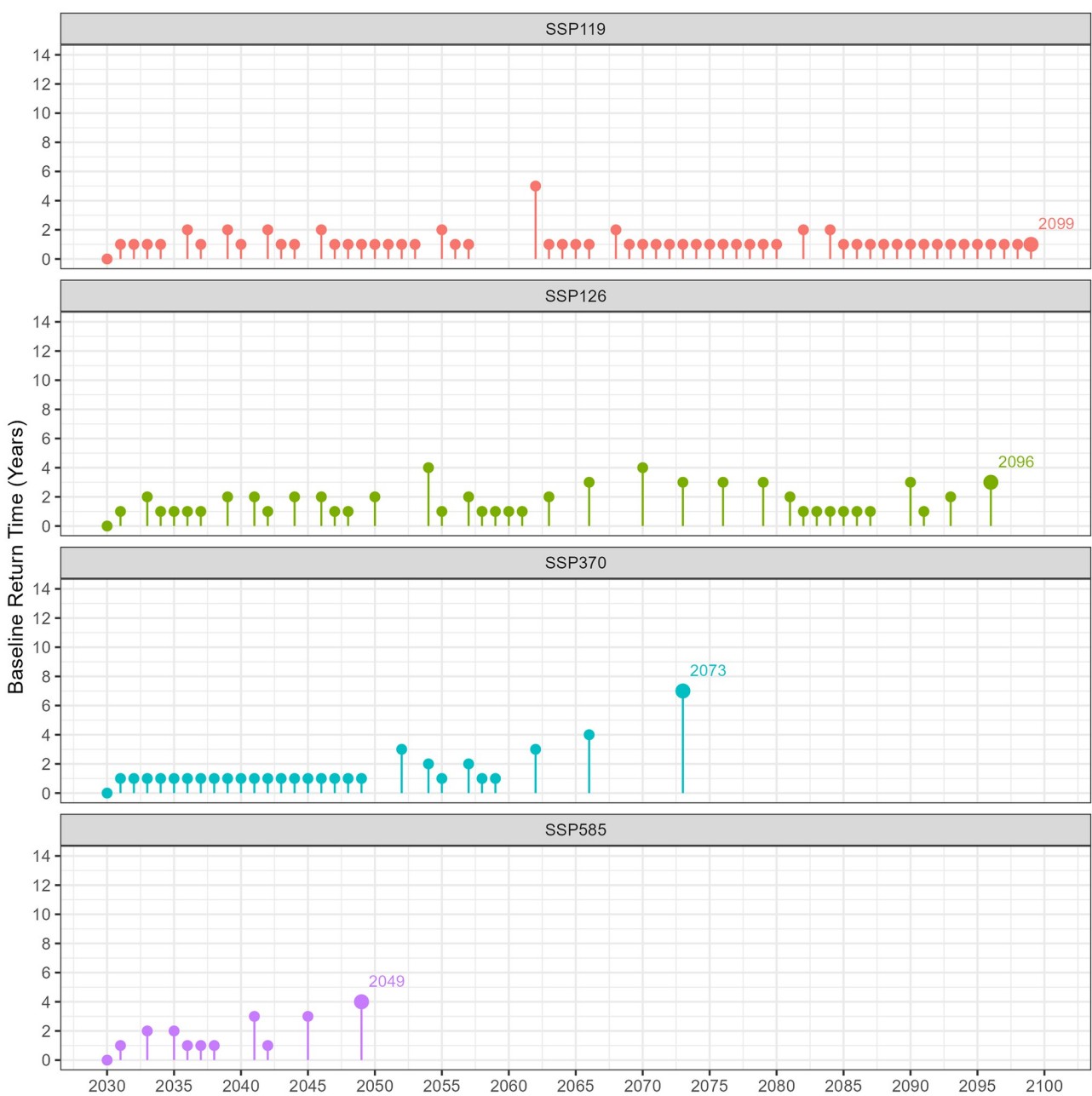

**Fig 7. Baseline return time for moderate shoot density ratios of *Z. muelleri* in Gladstone, Australia, depicting various socioeconomic pathway scenarios from 2030 to 2100.** Each colour represents a distinct scenario, as follows: red (SSP1-1.9), green (SSP1-2.6), blue (SSP3-7.0), and purple (SSP5-8.5). The lollipop graphs illustrate the years since the last event, where the moderate shoot density ratio was equal to or exceeded 0.9.

respect to a baseline year, focusing on the primary response variable for management, realized shoot density.

## 4.1 Annual changes in shoot density ratios

Annual projections for the seagrass shoot density ratio of *Zostera muelleri* in Gladstone are key for assessing its resilience, measuring immediate response and long-term persistence under

environmental stress, and understanding its health in changing climates. The mean shoot density ratio at distinct time points *t* provides insight into the expected trajectory of *Zostera muelleri*. The utilization of 50% and 90% PIs serves to elucidate the distribution of the ratio. When the bounds of these intervals are close, as illustrated in Fig 4 for 2033 under the SSP1-1.9 scenario, it indicates a high degree of certainty regarding a ratio approximating the mean. Conversely, a substantial separation between the bounds, as observed in Fig 4 for 2046 under the SSP1-1.9 scenario, denotes considerable uncertainty in those years. Uncertainty in ecological predictions, often stemming from limited information, underscores the need for intensified monitoring and additional research, especially during critical periods of uncertainty. These efforts are crucial for filling knowledge gaps and refining models, ultimately leading to more accurate and reliable ecological forecasting.

In the context of SSP1-1.9 and SSP1-2.6 scenarios—both of which project optimistic human development and a transition to sustainability—the annual high shoot density ratios for *Z. muelleri* exhibit considerable fluctuation over time. Despite this variability, the high shoot density ratio often rebounds, a trend that aligns well with the sustainability-oriented growth these scenarios anticipate and support with corresponding recovery time metrics. It is important to note that variability in shoot density is a common characteristic of seagrasses; thus, the observed fluctuations over time in the SSP1-1.9 and SSP1-2.6 scenarios align with this trait. Specifically, the pattern of decline and subsequent rebound in shoot density ratios has been historically documented in this region's dynamic tropical seagrass meadows. However, such fluctuation patterns might not be as prevalent in regions with more stable environmental conditions and steadier ecological trajectories, where seagrass meadows exhibit less variability.

Given the lower emissions and enhanced environmental stewardship under these pathways, conditions appear more favorable for *Z. muelleri* to thrive, corresponding to high resilience overall. In the SSP1-1.9 and SSP1-2.6 scenarios, significant fluctuations in the high shoot density ratio are observed, with a corresponding increase in uncertainty. Following this phase of high variability, the high shoot density ratio typically stabilizes to nominal levels, as exemplified by the years 2041 and 2052 in the SSP1-1.9 scenario, illustrated in Fig 4. For instance, in 2041, the high shoot density ratio was 0.2 (90% PI of 0.0042) but increased to 0.5 (90% PI of 0.8742) in 2044 (S1 Table). Over time, this trend indicates a decline, with an increasing certainty in its trajectory. A detailed examination of the Monte Carlo simulation samples reveals various heat stress event scenarios in the 2040s, which, through cumulative effects, contribute to significantly different outcomes in shoot density and resilience (S7 Fig). Additionally, the temperature optimality differed, further amplifying the impact on resilience.

Under the SSP1-2.6 scenario (Fig 4), the 2030s initially maintain baseline levels. However, a noticeable downtrend emerges from the mid-2040s through the 2060s, accompanied by increasing uncertainty. Within this scenario, carbon emissions are projected to decrease from current levels, achieving net zero by approximately 2075. Concurrently, temperatures are forecasted to rise significantly in the short term (2021-2040), increasing by roughly 1.5˚C. This warming trajectory is anticipated to persist, with temperatures escalating to 1.7˚C between 2041 and 2060 [135]. The marked decline in high population during the 2040s and 2060s can plausibly be attributed to the SSP1-26 projections. The heightened uncertainty may arise from variations among the samples, as depicted in S8 Fig, which presents three temperature projections for the 2050s.

Conversely, under the medium-high emissions and population growth scenario (SSP3-7.0), as well as the highest emissions pathway (SSP5-8.5), a notable decline in the high shoot density ratios of *Z. muelleri* is observed. Although the spread of values for these percentiles is initially broad, it narrows as the century progresses. For instance, in the year 2030 SSP5-8.5 scenario,

the interval between the $5^{th}$ and $95^{th}$ percentile is from 0.1365 to 0.4568, a range of 0.3203 PI (0.9) ($r_{high}(2030)$ = 0.3759) (S4 Table). However, by the year 2050, this range has narrowed significantly, with the $5^{th}$ percentile at 0.1307 and the $95^{th}$ percentile at 0.1344, resulting in a much smaller PI(0.9) of 0.0037 ($r_{high}(2050)$ = 0.1327). This trend of contraction continues such that by the year 2099, the PI(0.9) has reduced to less than 0.014 ($r_{high}(2099)$ = 0.0560), indicating a more precise prediction towards the end of the century. The projected trend indicates that by mid-century (2050) and end-of-century (2100), environmental conditions under both scenarios will be less favorable for seagrasses, underscoring the importance of conservation planning in the face of climate change [136]. Adverse effects of climate change on seagrasses, as foundational species, could lead to the disruption of biological relationships among other ecologically connected taxonomic groups [137, 138]. This may heighten their susceptibility to ecological disturbances and evolutionary shifts, culminating in potential site extinction [138, 139].

## 4.2 Post-disturbance recovery time

According to the Environmental Protection Agency of Western Australia, a meadow is considered to be a permanent loss if the meadow does not recover within five years [140]. In the context of high shoot density ratio recovery times, the SSP1-1.9 pathway presents the lowest risk, featuring an average recovery time of 2.44 years. Nonetheless, during the 2040s, a 12-year gap emerges where the meadow fails to recover, surpassing the 5-year threshold and thereby signaling a potential risk of meadow extinction under specific conditions. Although the meadow appears to recover until the mid-2040s, the SSP1-2.6 pathway presents a particularly concerning situation in the mid-2050s and 2070s, with peaks in recovery time lasting 12 and 18 years, respectively. If conditions continue to be favorable for the meadow through 2045 and mitigation measures are enacted at that point, the decline of seagrass might be slowed. Particularly concerning are the SSP3-7.0 and SSP5-8.5 scenario, with fewer recorded recovery events that persist until 2049 and 2038, respectively. Meadows under these scenarios face a significantly elevated risk of being classified as extinct. The results suggest that seagrass meadows could face varying degrees of site extinction risk depending on the SSP.

According to the Australian and New Zealand water quality guidelines [141], ecosystem management frequently employs a reference site for comparison. Specifically, the median of a managed site is evaluated against the $20^{th}$ and $80^{th}$ percentiles of a reference site(s) under the 'moderate protection' criterion. In accordance with [141], S5 and S6 Figs present the baseline return times for high and moderate shoot density ratios, respectively, employing a threshold of 0.8 rather than the conventional 0.9. Despite employing an adjusted threshold, the findings reveal that the most recent occurrences of high shoot density return times for the SSP3-7.0 and SSP5-8.5 scenarios were logged in the years 2055 and 2038, respectively. Post-2038, the prospect of returning to the baseline becomes untenable, culminating in an enduring diminution of meadow populations. This trend is corroborated by a continuous escalation in the zero shoot density ratio, as illustrated in Fig 5. Before 2038, the zero shoot density ratio consistently hovers at approximately 1. However, a pronounced and consistent increase in the zero ratio is observed in the following period. Conversely, for the SSP1-1.9 and SSP1-2.6 scenarios, a significant decrease in return times was observed, with heightened periods of risk specifically identified in the decades of the 2040s and 2060s.

## 4.3 Contributions and future research opportunities

Over the past century, there has been a significant increase in the duration and frequency of marine heatwaves worldwide, with average increases of 17% and 34%, respectively. This has

resulted in a concerning 54% rise in the annual number of marine heatwaves days [142]. Given that extreme climate events can induce significant ecological shifts [17], understanding the direction and magnitude of these shifts is imperative. Increasing occurrences of marine heatwaves, alongside their rising intensity and longer durations, have been extensively studied, particularly their anticipated escalation throughout the 21st century due to climate change [5, 143]. However, many studies overlook factors beyond marine heatwave frequency and intensity, failing to consider their broader impacts on marine ecosystems.

To address this gap, our study employs a DBN model known for its ability to integrate diverse data sources into a dynamic, whole-of-system model [144, 145]. The primary contribution of this research lies in the novel integration of the DBN with climate models. We utilized an ensemble of downscaled climate models specifically tailored for coastal areas. This ensemble approach is expected to surpass the performance of individual model runs. This approach was enhanced by integrating various SSP to represent different socio-economic futures. Additionally, we developed a framework to translate mean temperature data into metrics aligned with the thermal optimality for *Z. muelleri*. The study also advances the methodology by applying Monte Carlo simulations and introducing novel resilience metrics, thereby creating a transparent and explainable model. While current metrics effectively evaluate resilience across various scenarios, future research could benefit from examining additional metrics to enhance our understanding of seagrass resilience, particularly under more challenging conditions.

The DBN presented provides a versatile tool for risk assessment that can be continuously refined with new data, including observational and field data or updated model results, ensuring the ongoing relevance and accuracy of the risk evaluations. The model design is not static; it can be adapted to accommodate for new climate scenarios, applied to different regions, or tailored to assess risks to various species beyond seagrass. To adapt the model, key steps include downscaling the climate model for regional relevance, analyzing the relationship between high-resolution temperature data and climate model data to apply correct thresholds, and identifying the optimal temperature thresholds for targeted species, including the critical temperature levels and durations that lead to shoot mortality. As scientific understanding evolves, the DBN can be iteratively adjusted by integrating additional variables to capture emerging characteristics or removing those no longer pertinent. This study highlights numerous avenues for future research, particularly in identifying model parameters vulnerable to uncertainty. Upon determining these parameters, probability distributions can be assigned to each, reflecting the range of possible values. Expert judgment, empirical data, or an exhaustive literature review can guide the choice of these distributions.

In this study, Monte Carlo simulations were employed to incorporate the uncertainty of climate projections into a DBN. It has been suggested that Markov Chain Monte Carlo methods could be utilized for inference within a DBN to capture the uncertainty in both model parameters and climate projections explicitly [146]. However, this approach presents significant computational challenges and is considered an avenue for future research. The methodologies applied in this study were conservative yet adequate for the scope of our research. In addition, a more comprehensive study on forecasting methods for temperature and other environmental impacts on ecosystems represents another potential area for future research.

Furthermore, conducting sensitivity analyses would provide invaluable insights into the robustness of the model, particularly in how thresholds are tested and validated. These analyses aim to identify the parameters that most significantly impact overall risk estimates. Several methods could be employed for this purpose, including calculating the partial derivatives of the risk estimate with respect to each parameter or utilizing advanced techniques like variance-based sensitivity analysis. Incorporating expert knowledge in these analyses is crucial, as it allows for the refinement of thresholds based on practical experience and insights that are

not always captured by empirical data alone. This integration enhances the predictive reliability of the model and guides future data collection and research efforts, ultimately contributing to a more robust and adaptable decision-support tool for marine ecosystem management. Future work could also extend the research to capture how different heatwave frequencies and durations more comprehensively, and their variations over time might arise under different SSPs and their impact on resilience.

Additionally, the current study did not explore variations in thresholds for estimating temperature and heat stress state probabilities, akin to scenario assessments. Future research should consider how adjustments in these thresholds, informed by expert knowledge and sensitivity analysis outcomes, might influence model results. Such investigations would bridge identified gaps and limitations, offering new opportunities for more detailed and comprehensive findings in marine ecosystem management. These efforts are essential to ensure that the model remains adaptable and responsive to emerging data and expert consensus, thereby improving its utility as a decision-support tool.

## 5 Conclusion

For the conservation and survival of seagrass species, it is imperative that management strategies are both flexible and adaptive in response to the uncertainties inherent in climate change projections. The methodology employed in this study facilitates the identification of resilience outcomes—ratios and recovery times—along with their fluctuations and associated uncertainties. This study provides significant insights into the resilience of *Z. muelleri* under diverse SSP scenarios, highlighting periods of high uncertainty that necessitate immediate action through intensified monitoring or further research to mitigate potential risks. Determining that SSP1-1.9 and SSP1-2.6 enable the best chance of recovery under future warming. The methods and approaches outlined are versatile and adaptable, making them suitable for application across a broad range of species, sites, climate scenarios, and ecosystems.

## Supporting information

**S1 Fig. Seagrass Dynamic Bayesian Network.** A Bayesian Network (BN) is a class of graphical models representing the probabilistic relationships among a set of variables (or nodes) with directed arcs between them. These variables form a directed acyclic graph (DAG), and each node in the DAG has a state that varies depending on the states of other nodes [147]. Information about those states is transmitted throughout the DAG, and as a result, inferences may be made by adding new data or evidence to the network. While BNs are suitable approaches for inferring static processes, the Dynamic Bayesian Network (DBN), a BN extension in which nodes represent variables at specific time slices, may be used to model temporally varying processes in which the state of a variable may change over time [148]. When adding a dynamic component to a BN and creating a DBN, the characteristic processes of complex systems, including cumulative effects and feedback processes, can be captured. The DBN approach allows us to predict the resilience of a system given the temporal dynamics of the components of the ecosystem and their interactions with natural and anthropogenic stressors [101, 107]. The overall DBN network structure is depicted with ovals representing factors (or nodes) and arrows indicating causal parent-child relationships. In this structure, a parent node (e.g., Meadow Type) exerts an influence on a child node (e.g., Location Type). Conversely, the absence of a link between nodes signifies conditional independence. Rounded rectangles denote subnetworks [77]. Note the presence of complex interdependencies and feedback loops in the seagrass ecosystem. Node colors denote different categories: site condition (white), recovery (purple), resistance (green), environmental (blue), and population (yellow). The

following symbology is used in this figure: A node with a check mark: A node is ticked when inference has been successfully executed. A curved arrow back onto itself: Denotes a link to the node's next time slice (i.e., $t + 1$). A double-headed arrow: Indicates that arcs are heading both ways between two subnetworks. For example, a double-headed arrow between a node N and a subnetwork S means that there is at least one node in S that depends on N and that there is at least one node in S that influences N. An arrow labeled with a [1]: Indicates a connection to a subsequent time slice ($t + 1$).
(PDF)

**S2 Fig. Annual projections for moderate shoot density ratio of *Z. muelleri* in Gladstone, Australia, depicting different socio-economic pathway scenarios from 2030 to 2100.** Each colour corresponds to a distinct scenario: red (SSP1-1.9), green (SSP1-2.6), blue (SSP3-7.0), and purple (SSP5-8.5). The lines represent key statistical measures: solid lines for the average, dashed lines for the 95$^{th}$ percentile (upper) and 5$^{th}$ percentile (lower), a grey area for the 75$^{th}$ percentile (upper) and 25$^{th}$ percentile (lower) derived from a dataset of 100 samples.
(PDF)

**S3 Fig. Annual projections for low shoot density ratio of *Z. muelleri* in Gladstone, Australia, depicting different socio-economic pathway scenarios from 2030 to 2100.** Each colour corresponds to a distinct scenario: red (SSP1-1.9), green (SSP1-2.6), blue (SSP3-7.0), and purple (SSP5-8.5). The lines represent key statistical measures: solid lines for the average, dashed lines for the 95$^{th}$ percentile (upper) and 5$^{th}$ percentile (lower), a grey area for the 75$^{th}$ percentile (upper) and 25$^{th}$ percentile (lower) derived from a dataset of 100 samples.
(PDF)

**S4 Fig. Recovery time.** The estimation of recovery time $q(y)$ is calculated as $q(y) = y' - y$, where $y$ is the first year when the high ratio $r_{\text{high}}(y)$ falls below 90% of the baseline. Then $y'$ is the earliest subsequent year when the high ratio $r_{\text{high}}(y')$ returns to or exceeds 90% of the baseline.
(PDF)

**S5 Fig. Baseline return time for high shoot density ratios of *Z. muelleri* in Gladstone, Australia, depicting various socio-economic pathway scenarios from 2030 to 2100.** Each color represents a distinct scenario, as follows: red (SSP1-1.9), green (SSP1-2.6), blue (SSP3-7.0), and purple (SSP5-8.5). The lollipop graphs illustrate the number of years since the last event where the high shoot density ratio was equal to or exceeded 0.8.
(PDF)

**S6 Fig. Baseline return time for moderate shoot density ratios of *Z. muelleri* in Gladstone, Australia, depicting various socio-economic pathway scenarios from 2030 to 2100.** Each color represents a distinct scenario, as follows: red (SSP1-1.9), green (SSP1-2.6), blue (SSP3-7.0), and purple (SSP5-8.5). The lollipop graphs illustrate the number of years since the last event where the moderate shoot density ratio was equal to or exceeded 0.8.
(PDF)

**S7 Fig. Predicted-state probabilities for high shoot density and the effect of heat stress on *Z. muelleri* in Gladstone, Australia, under the SSP1-1.9 scenario.** (a) The green line represents the baseline high shoot density, and the purple lines represent the projected high shoot density. (b) The green line illustrates the baseline effect of heat stress, while the red lines depict the projected impact of heat stress. No baseline is represented for heat stress, as its value is consistently zero. (c) The green line portrays the baseline temperature effect, and the orange lines

indicate the projected sub-optimal temperatures.
(PDF)

**S8 Fig. Predicted-state probabilities for high shoot density and the effect of heat stress on *Z. muelleri* in Gladstone, Australia, under the SSP1-2.6 scenario. (a)** The green line represents the baseline high shoot density, and the purple lines represent the projected high shoot density. **(b)** The green line illustrates the baseline effect of heat stress, while the red lines depict the projected impact of heat stress. No baseline is represented for heat stress, as its value is consistently zero. **(c)** The green line portrays the baseline temperature effect, and the orange lines indicate the projected sub-optimal temperatures.
(PDF)

**S1 Table. High shoot density ratio across years for SSP1-1.9 scenario: This table provides an analysis of the high shoot density states, measured annually within the SSP1-1.9 scenario. Avg:** denotes the average high shoot density ratio per decade. **Q25:** represents $25^{th}$ percentile, marking the value below which 25% of the observations fall. **Q95:** stands for the $95^{th}$ percentile indicating the value below which 95% of the observations are found.
(PDF)

**S2 Table. High shoot density ratio across years for SSP1-2.6 scenario: This table provides an analysis of the high shoot density states, measured annually within the SSP1-2.6 scenario. Avg:** denotes the average high shoot density ratio per decade. **Q25:** represents $25^{th}$ percentile, marking the value below which 25% of the observations fall. **Q95:** stands for the $95^{th}$ percentile indicating the value below which 95% of the observations are found.
(PDF)

**S3 Table. High shoot density ratio across years for SSP3-7.0 scenario: This table provides an analysis of the high shoot density states, measured annually within the SSP3-7.0 scenario. Avg:** denotes the average high shoot density ratio per decade. **Q25:** represents $25^{th}$ percentile, marking the value below which 25% of the observations fall. **Q95:** stands for the $95^{th}$ percentile indicating the value below which 95% of the observations are found.
(PDF)

**S4 Table. High shoot density ratio across years for SSP5-8.5 scenario: This table provides an analysis of the high shoot density states, measured annually within the SSP5-8.5 scenario. Avg:** denotes the average high shoot density ratio per decade. **Q25:** represents $25^{th}$ percentile, marking the value below which 25% of the observations fall. **Q95:** stands for the $95^{th}$ percentile indicating the value below which 95% of the observations are found.
(PDF)

**S5 Table. Moderate shoot density ratio across years for SSP1-1.9 scenario: This table provides an analysis of the moderate shoot density states, measured annually within the SSP1-1.9 scenario. Avg:** denotes the average moderate shoot density ratio per decade. **Q25:** represents $25^{th}$ percentile, marking the value below which 25% of the observations fall. **Q95:** stands for the $95^{th}$ percentile indicating the value below which 95% of the observations are found.
(PDF)

**S6 Table. Moderate shoot density ratio across years for SSP1-2.6 scenario: This table provides an analysis of the moderate shoot density states, measured annually within the SSP1-2.6 scenario. Avg:** denotes the average moderate shoot density ratio per decade. **Q25:** represents $25^{th}$ percentile, marking the value below which 25% of the observations fall. **Q95:** stands

for the 95<sup>th</sup> percentile indicating the value below which 95% of the observations are found.
(PDF)

**S7 Table. Moderate shoot density ratio across years for SSP3-7.0 scenario: This table provides an analysis of the moderate shoot density states, measured annually within the SSP3-7.0 scenario. Avg:** denotes the average moderate shoot density ratio per decade. **Q25:** represents 25<sup>th</sup> percentile, marking the value below which 25% of the observations fall. **Q95:** stands for the 95<sup>th</sup> percentile indicating the value below which 95% of the observations are found.
(PDF)

**S8 Table. Moderate shoot density ratio across years for SSP5-8.5 scenario: This table provides an analysis of the moderate shoot density states, measured annually within the SSP5-8.5 scenario. Avg:** denotes the average moderate shoot density ratio per decade. **Q25:** represents 25<sup>th</sup> percentile, marking the value below which 25% of the observations fall. **Q95:** stands for the 95<sup>th</sup> percentile indicating the value below which 95% of the observations are found.
(PDF)

**S9 Table. Low shoot density ratio across years for SSP1-1.9 scenario: This table provides an analysis of the low shoot density states, measured annually within the SSP1-1.9 scenario. Avg:** denotes the average low shoot density ratio per decade. **Q25:** represents 25<sup>th</sup> percentile, marking the value below which 25% of the observations fall. **Q95:** stands for the 95<sup>th</sup> percentile indicating the value below which 95% of the observations are found.
(PDF)

**S10 Table. Low shoot density ratio across years for SSP1-2.6 scenario: This table provides an analysis of the low shoot density states, measured annually within the SSP1-2.6 scenario. Avg:** denotes the average low shoot density ratio per decade. **Q25:** represents 25<sup>th</sup> percentile, marking the value below which 25% of the observations fall. **Q95:** stands for the 95<sup>th</sup> percentile indicating the value below which 95% of the observations are found.
(PDF)

**S11 Table. Low shoot density ratio across years for SSP3-7.0 scenario: This table provides an analysis of the low shoot density states, measured annually within the SSP3-7.0 scenario. Avg:** denotes the average low shoot density ratio per decade. **Q25:** represents 25<sup>th</sup> percentile, marking the value below which 25% of the observations fall. **Q95:** stands for the 95<sup>th</sup> percentile indicating the value below which 95% of the observations are found.
(PDF)

**S12 Table. Low shoot density ratio across years for SSP5-8.5 scenario: This table provides an analysis of the low shoot density states, measured annually within the SSP5-8.5 scenario. Avg:** denotes the average low shoot density ratio per decade. **Q25:** represents 25<sup>th</sup> percentile, marking the value below which 25% of the observations fall. **Q95:** stands for the 95<sup>th</sup> percentile indicating the value below which 95% of the observations are found.
(PDF)

**S13 Table. Zero shoot density ratio across years for SSP1-1.9 scenario: This table provides an analysis of the zero shoot density states, measured annually within the SSP1-1.9 scenario. Avg:** denotes the average zero shoot density ratio per decade. **Q25:** represents 25<sup>th</sup> percentile, marking the value below which 25% of the observations fall. **Q95:** stands for the 95<sup>th</sup> percentile indicating the value below which 95% of the observations are found.
(PDF)

**S14 Table. Zero shoot density ratio across years for SSP1-2.6 scenario: This table provides an analysis of the zero shoot density states, measured annually within the SSP1-2.6 scenario. Avg:** denotes the average zero shoot density ratio per decade. **Q25:** represents $25^{th}$ percentile, marking the value below which 25% of the observations fall. **Q95:** stands for the $95^{th}$ percentile indicating the value below which 95% of the observations are found.
(PDF)

**S15 Table. Zero shoot density ratio across years for SSP3-7.0 scenario: This table provides an analysis of the zero shoot density states, measured annually within the SSP3-7.0 scenario. Avg:** denotes the average zero shoot density ratio per decade. **Q25:** represents $25^{th}$ percentile, marking the value below which 25% of the observations fall. **Q95:** stands for the $95^{th}$ percentile indicating the value below which 95% of the observations are found.
(PDF)

**S16 Table. Zero shoot density ratio across years for SSP5-8.5 scenario: This table provides an analysis of the zero shoot density states, measured annually within the SSP5-8.5 scenario. Avg:** denotes the average zero shoot density ratio per decade. **Q25:** represents $25^{th}$ percentile, marking the value below which 25% of the observations fall. **Q95:** stands for the $95^{th}$ percentile indicating the value below which 95% of the observations are found.
(PDF)

## Acknowledgments

The author thanks the Centre for Data Science (CDS) for its invaluable support and guidance throughout this research. The authors also thank the Queensland University of Technology (QUT) for providing logistical and infrastructural support.

## Author Contributions

**Conceptualization:** Kathryn McMahon, Paul P.-Y. Wu.

**Data curation:** Jennifer K. McWhorter.

**Formal analysis:** Paula S. Hatum, Paul P.-Y. Wu.

**Investigation:** Paula S. Hatum, Paul P.-Y. Wu.

**Methodology:** Paula S. Hatum, Kathryn McMahon, Kerrie Mengersen, Paul P.-Y. Wu.

**Supervision:** Kathryn McMahon, Kerrie Mengersen, Paul P.-Y. Wu.

**Validation:** Paula S. Hatum, Kathryn McMahon, Paul P.-Y. Wu.

**Visualization:** Paula S. Hatum, Kerrie Mengersen, Paul P.-Y. Wu.

**Writing – original draft:** Paula S. Hatum.

**Writing – review & editing:** Kathryn McMahon, Kerrie Mengersen, Jennifer K. McWhorter, Paul P.-Y. Wu.

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
