## [Decision Letter · Decision Letter 0]

29 May 2024

PONE-D-24-04137In hot water: uncertainties in projecting marine heatwaves impacts on seagrass meadowsPLOS ONE

Dear Dr. Hatum,i

Thank you for submitting your manuscript to PLOS ONE. After careful consideration, we feel that it has merit but does not fully meet PLOS ONE’s publication criteria as it currently stands. Therefore, we invite you to submit a revised version of the manuscript that fully addresses all the points raised during the review process by both reviewers.

We look forward to receiving your revised manuscript.

Kind regards,

João Miguel Dias, Ph.D.

Academic Editor

PLOS ONE

Journal Requirements:

4. Thank you for stating the following financial disclosure: "This work was supported by the Centre of Data Science (CDS) and by QUT South American Scholarships." 

5. Thank you for stating the following in the Acknowledgments Section of your manuscript: ' The authors also thank the Queensland University 868

of Technology (QUT) for the financial support received through the South American 869

Scholarship."

Please remove any funding-related text from the manuscript and let us know how you would like to update your Funding Statement. Currently, your Funding Statement reads as follows: "This work was supported by the Centre of Data Science (CDS) and by QUT South American Scholarships." 

6. We note that Figure 1 in your submission contain [map/satellite] images which may be copyrighted. All PLOS content is published under the Creative Commons Attribution License (CC BY 4.0), which means that the manuscript, images, and Supporting Information files will be freely available online, and any third party is permitted to access, download, copy, distribute, and use these materials in any way, even commercially, with proper attribution. For these reasons, we cannot publish previously copyrighted maps or satellite images created using proprietary data, such as Google software (Google Maps, Street View, and Earth). For more information, see our copyright guidelines: http://journals.plos.org/plosone/s/licenses-and-copyright.

7. We notice that your supplementary tables are included in the manuscript file. Please remove them and upload them with the file type 'Supporting Information'. Please ensure that each Supporting Information file has a legend listed in the manuscript after the references list.

Reviewers' comments:

Reviewer's Responses to Questions

**Comments to the Author**

1. Is the manuscript technically sound, and do the data support the conclusions?

Reviewer #1: Yes

Reviewer #2: Partly

2. Has the statistical analysis been performed appropriately and rigorously? 

Reviewer #1: Yes

Reviewer #2: I Don't Know

3. Have the authors made all data underlying the findings in their manuscript fully available?

Reviewer #1: Yes

Reviewer #2: Yes

4. Is the manuscript presented in an intelligible fashion and written in standard English?

Reviewer #1: Yes

Reviewer #2: Yes

5. Review Comments to the Author

Reviewer #1: In my opinion, the methodology presented here is powerful and promising to support risk analysis associated to predictions for marine ecosystems under climate change.

I thoroughly enjoyed reading this manuscript as it is exceptionally well-written and presents a robust framework. In addition to the few points below, I suggest condensing the Discussion section slightly, although I acknowledge this may pose a challenge given the numerous projected scenarios.

Except for some minor points (below), in my opinion, I recommend this manuscript for publication in PLOS ONE

Pg. 5, line 141: change to meadow

Pg. 8; lines 267-271- I wonder about the assumption of the 30 ºC threshold for shoot mortality, which seems to be only relying on data from 103. Wouldn’t it be better to consider more studies with seagrasses from tropical areas (i.e. similar conditions to Gladstone)? Can the authors, please, comment on this point?

Pg. 8, line 274: replace “ecosystem DBN” by either “framework” or “model” DBN

Pg. 10, lines 336-338: I suppose there isn’t previous data on shoot density, from the 1990’s or early 2000’s or before; if so, it would be preferable to assume this data as baseline. Please, comment.

Pg. 11 and 12, lines 375-377: reformulate the two sentences starting by “Post-integration of heat stress nodes”.

Pg. 13, line 443: change to there is; also change throughout the manuscript

Pg. 21- There’s a repetition of the Conclusion section

Figures- Improve the quality and definition of graphs and Figures

Reviewer #2: General comment

This study aimed to develop a methodological framework that integrates ecosystem DBNs with climate model projections to assess the resilience of seagrass ecosystems, and focuses on enhancing our understanding of seagrass responses to marine heat waves. The paper is well written, but there are some aspects that should be improved to clarify the understanding of the readers. First of all, it is necessary to clarify how marine heatwaves are considered in this study. In the methods and results, the authors describe that they use SST data from CMIP6 models, but the number of marine heatwave events for the different scenarios and the duration are not assessed. This leaves me wondering whether you analyzed marine heatwaves or simply SST variations over time. I have another question regarding mean sea level variations. Given that seagrasses are highly sensitive to changes in mean sea level, I wonder how this effect is accounted for in your model and in this study. Since several studies indicate that mean sea level is rising on the Australian coast and will continue to rise until 2100, it is crucial to clarify how this effect could impact seagrasses. Finally, it is crucial to clarify details about the model and its validation. I suggest adding a description of the model as supplementary material and clarifying how the model was validated.

Specific comments

(139) Study area: Include more information. What about tides, waves, sea levels and other factors (e.g. human induced) affecting seagrasses evolution? How study area seagrasses have evolved over time? Are there evidences that marine heatwaves affected its evolution? If yes, report the episodes. Fig. 1 – Provide more details of the study area, including pictures of the seagrasses. Include labels with locations referred in the text: Gladstone Harbour; Pelican Banks; Great Barrier ReefWorld Heritage Area (GBR); Australia

(line 152) Seagrass Dynamic Bayesian Network model

Consider including a significant portion of this information as supplementary material. For example, lines 153 to 165 contain a description of the model rather than a description of the methods.

Consider shortening fig. 2 caption. Journal recommendations state: avoid lengthy descriptions of methods.

(213-216) Provide a detailed explanation of how you downloaded the CMIP6 data and describe in detail how you downscaled the data to a 10 km resolution.

6. PLOS authors have the option to publish the peer review history of their article (what does this mean?). If published, this will include your full peer review and any attached files.

Reviewer #1: **Yes: **Irene Martins

Reviewer #2: No

---

## [Author Response · Author response to Decision Letter 0]

17 Jul 2024

Journal Requirements:

The PLOS ONE style templates can be found at:

https://journals.plos.org/plosone/s/file?id=wjVg/PLOSOne_formatting_sample_main_body.pdf and, 

4. Thank you for stating the following financial disclosure: "This work was supported by the Centre of Data Science (CDS) and by QUT South American Scholarships." Please state what role the funders took in the study. If the funders had no role, please state: "The funders had no role in study design, data collection and analysis, decision to publish, or preparation of the manuscript." If this statement is not correct you must amend it as needed. Please include this amended Role of Funder statement in your cover letter; we will change the online submission form on your behalf.

5. Thank you for stating the following in the Acknowledgments Section of your manuscript: ' The authors also thank the Queensland University of Technology (QUT) for the financial support received through the South American Scholarship." We note that you have provided funding information that is not currently declared in your Funding Statement. However, funding information should not appear in the Acknowledgments section or other areas of your manuscript. We will only publish funding information present in the Funding Statement section of the online submission form. Please remove any funding-related text from the manuscript and let us know how you would like to update your Funding Statement. Currently, your Funding Statement reads as follows: "This work was supported by the Centre of Data Science (CDS) and by QUT South American Scholarships." Please include your amended statements within your cover letter; we will change the online submission form on your behalf.

6. We note that Figure 1 in your submission contain [map/satellite] images which may be copyrighted. All PLOS content is published under the Creative Commons Attribution License (CC BY 4.0), which means that the manuscript, images, and Supporting Information files will be freely available online, and any third party is permitted to access, download, copy, distribute, and use these materials in any way, even commercially, with proper attribution. For these reasons, we cannot publish previously copyrighted maps or satellite images created using proprietary data, such as Google software (Google Maps, Street View, and Earth). For more information, see our copyright guidelines: 

http://journals.plos.org/plosone/s/licenses-and-copyright.

a. You may seek permission from the original copyright holder of Figure 1 to publish the content specifically under the CC BY 4.0 license. We recommend that you contact the original copyright holder with the Content Permission Form (http://journals.plos.org/plosone/s/file?id=7c09/content-permission-form.pdf) and the following text: “I request permission for the open-access journal PLOS ONE to publish XXX under the Creative Commons Attribution License (CCAL) CC BY 4.0 (http://creativecommons.org/licenses/by/4.0/). Please be aware that this license allows unrestricted use and distribution, even commercially, by third parties. Please reply and provide explicit written permission to publish XXX under a CC BY license and complete the attached form.”

Please upload the completed Content Permission Form or other proof of granted permissions as an ""Other"" file with your submission. In the figure caption of the copyrighted figure, please include the following text: “Reprinted from [ref] under a CC BY license, with permission from [name of publisher], original copyright [original Copyright year].”

b. If you are unable to obtain permission from the original copyright holder to publish these figures under the CC BY 4.0 license or if the copyright holder’s requirements are incompatible with the CC BY 4.0 license, please either i) remove the figure or ii) supply a replacement figure that complies with the CC BY 4.0 license. Please check copyright information on all replacement figures and update the figure caption with source information. If applicable, please specify in the figure caption text when a figure is similar but not identical to the original image and is therefore for illustrative purposes only. The following resources for replacing copyrighted map figures may be helpful:

USGS National Map Viewer (public domain): 

http://viewer.nationalmap.gov/viewer/

The Gateway to Astronaut Photography of Earth (public domain): 

http://eol.jsc.nasa.gov/sseop/clickmap/

Maps at the CIA (public domain): 

https://www.cia.gov/library/publications/the-worldfactbook/index.html and 

https://www.cia.gov/library/publications/cia-maps-publications/index.html

NASA Earth Observatory (public domain): 

http://earthobservatory.nasa.gov/

Landsat: 

http://landsat.visibleearth.nasa.gov/

USGS EROS (Earth Resources Observatory and Science (EROS) Center) (public domain): 

http://eros.usgs.gov/#

Natural Earth (public domain): 

http://www.naturalearthdata.com/

7. We notice that your supplementary tables are included in the manuscript file. Please remove them and upload them with the file type 'Supporting Information'. Please ensure that each Supporting Information file has a legend listed in the manuscript after the references list.

Authors Response:

Thank you for your detailed feedback on our manuscript submission. We have addressed the points as follows:

1. We have ensured that our manuscript meets PLOS ONE's style requirements, including proper file naming. We have utilised the provided PLOS ONE style templates for the main body and title, authors, and affiliations sections.

2. We have amended the financial disclosure statement to include the role of the funders. The revised statement reads: "The funders had no role in study design, data collection and analysis, decision to publish, or manuscript preparation." We have also removed the funding-related text from the Acknowledgments section of our manuscript. The updated Funding Statement now reads: "This work was supported by the Centre of Data Science (CDS) and by QUT South American Scholarships." This amended statement was included in the online submission form update cover letter.

3. Figure 1 Copyright Issue: 

• World Heritage Areas of Queensland: The data for World Heritage Areas of Queensland was obtained from the Queensland Government Spatial Catalogue, available at

https://qldspatial.information.qld.gov.au/catalogue/custom/detail.page?fid={A49A0B80-277C-40DC-AFB5-D95E1EF9959B}

This dataset is provided under the Queensland Government’s open data initiative, which supports the distribution and use of its spatial data under the Creative Commons Attribution 4.0 International (CC BY 4.0) license.

• State controlled roads – Queensland: The data for State Controlled Roads in Queensland was sourced from the same Queensland Government Spatial Catalogue, accessible at

https://qldspatial.information.qld.gov.au/catalogue/custom/detail.page?fid={6BD377F3-B007-437B-A523-D7B26C07F545}

This dataset is also covered under the CC BY 4.0 license, allowing for free use and distribution with proper attribution.

• States and Territories - 2021 (Shapefile); Local Government Areas - 2021 (Shapefile), and Suburbs and Localities – 2021:

https://www.abs.gov.au/statistics/standards/australian-statistical-geography-standard-asgs-edition-3/jul2021-jun2026/access-and-downloads/digital-boundary-files#metadata-for-digital-boundary-files

Reference: Australian Bureau of Statistics (Jul2021-Jun2026), Access and downloads, ABS Website, accessed 17 July 2024.

4. We have removed the supplementary tables from the manuscript file and uploaded them separately with the file type 'Supporting Information.' Each Supporting Information file now includes a legend listed in the manuscript after the references list.

Review Comments to the Author

Please use the space provided to explain your answers to the questions above. You may also include additional comments for the author, including concerns about dual publication, research ethics, or publication ethics. (Please upload your review as an attachment if it exceeds 20,000 characters).

Reviewer #1: 

In my opinion, the methodology presented here is powerful and promising to support risk analysis associated to predictions for marine ecosystems under climate change. I thoroughly enjoyed reading this manuscript as it is exceptionally well-written and presents a robust framework. In addition to the few points below, I suggest condensing the Discussion section slightly, although I acknowledge this may pose a challenge given the numerous projected scenarios. Except for some minor points (below), in my opinion, I recommend this manuscript for publication in PLOS ONE.

Authors Response:

Thank you for your thorough review and valuable feedback on our manuscript. We are pleased to hear that you found our methodology promising and the manuscript well-written. In response to your suggestion, we have made slight changes to the Discussion section for enhanced clarity.

Lines 741-744

Discussion:

We have rewritten “Conversely, the SSP3-7.0 and SSP5-8.5 scenarios, characterized by more pessimistic projections, reveal a marked decline in high shoot density ratios. SSP3-7.0 depicts a world fragmented by renewed nationalism, while SSP5-8.5 foresees unfettered, rapid growth. This decline in shoot density ratios can likely be attributed to increased emissions and less sustainable practices, adversely affecting Z. muelleri. Although the value spread for these percentiles is initially broad, it narrows as the century progresses.” as “Conversely, under the medium-high emissions and population growth scenario (SSP3-7.0), as well as the highest emissions pathway (SSP5-8.5), a notable decline in the high shoot density ratios of Z. muelleri is observed. Although the spread of values for these percentiles is initially broad, it narrows as the century progresses.” 

Lines 813-816

Discussion:

We have rewritten “This method effectively characterizes uncertainties, greatly enhancing the model's value in ecological forecasting.” to “While current metrics effectively evaluate resilience across various scenarios, future research could benefit from examining additional metrics to enhance our understanding of seagrass resilience, particularly under more challenging conditions.”

Lines 349-363

Material and Methods:

We have moved the paragraph below to the "Measurement and Projections of Temperature Data" section in the Materials and Methods. 

“Given the unpredictable nature of future human behavior, these scenarios present a spectrum of plausible climate futures, reflecting the outcomes of diverse policy choices. SSP1 advocates for reduced emissions and a shift towards renewable energy, promoting a sustainable future path [115]. In contrast, SSP3, which has a higher radiative forcing, suggests a future marked by increased pollution and limited progress in health and education [116]. SSP5 envisions a heavy reliance on fossil fuels, resulting in significant increases in greenhouse gas emissions [117]. It is important that these RCP scenarios are not predictive and, therefore, do not carry associated probabilities [118]. Their primary purpose is to provide decision-makers with a range of potential outcomes derived from various plausible choices, thereby facilitating informed decision-making. As an example of how these projections can be used, consider the variability observed in the shoot density ratios of Z. muelleri under different SSP320 scenarios. This variability serves as a vital indicator, guiding strategic resource allocation in times of uncertainty and underscoring the importance of prioritising interventions in regions most likely to experience significant impacts.”

Pg. 5, line 141: change to meadow

Authors Response:

We have revised "meadow" as suggested.

Line 189

“Our case study includes a Zostera muelleri seagrass meadow located at Pelican Banks, Gladstone Harbour (latitude: 23◦45′58′′S and longitude: 151◦18′30′′E), Australia (Fig 1).”

Pg. 8; lines 267-271- I wonder about the assumption of the 30 ºC threshold for shoot mortality, which seems to be only relying on data from 103. Wouldn’t it be better to consider more studies with seagrasses from tropical areas (i.e. similar conditions to Gladstone)? Can the authors, please, comment on this point?

Authors Response:

Our thresholds were informed by studies on Zostera muelleri conducted in regions such as Cockle Bay and Picnic Bay on Magnetic Island, and Moreton Bay near the northern Great Barrier Reef — areas close to Gladstone.

References:

Collier, C. J., Ow, Y. X., Langlois, L., Uthicke, S., Johansson, C. L., O'Brien, K. R., ... & Adams, M. P. (2017). Optimum temperatures for net primary productivity of three tropical seagrass species. Frontiers in Plant Science, 8, 1446.

Collier, C. J., Uthicke, S., & Waycott, M. (2011). Thermal tolerance of two seagrass species at contrasting light levels: implications for future distribution in the Great Barrier Reef. Limnology and Oceanography, 56(6), 2200-2210.

We acknowledge the limitations of relying solely on these studies, particularly given the absence of direct data from Gladstone. To address this, we incorporated expert opinions when setting these parameters. We adjusted the threshold for seagrass shoot mortality from 33°C to 30°C based on literature reviews and climate projections that suggested daily averages below these thresholds. This decision considers that daily averages may not accurately reflect peak temperatures, which cause heat stress. Recognizing the limitation of basing this threshold on a single study, we agree that incorporating data from additional studies, especially from tropical regions like Gladstone, would strengthen our model. Furthermore, exploring different thresholds could provide deeper insights; however, such analysis is beyond the scope of our current work. Our primary objective was to develop a methodology that integrates climate projections with dynamic Bayesian network models to assess impacts on marine ecosystems. The tool we developed has the capability to simulate and assess different thresholds, which can be explored in future work.

We have revised the methods section to include more clarification on how/why we chose this method. Additionally, in our discussion, we revisit this topic as a caveat to our study and place for future improvement and text is also included below. 

Methods section

Lines 373-378

“The temperature thresholds used in the analysis were particularly informed by research conducted on Zostera muelleri in locations such as Cockle Bay and Picnic Bay on Magnetic Island, and Moreton Bay near the northern Great Barrier Reef—areas geographically proximate to Gladstone. Recognizing the limitations stemming from relying on data exclusively from these areas, especially given the absence of direct data from Gladstone itself, the thresholds were refined and validated through expert opinions.”

Discussion section

Lin

---

## [Decision Letter · Decision Letter 1]

15 Aug 2024

PONE-D-24-04137R1In hot water: uncertainties in projecting marine heatwaves impacts on seagrass meadowsPLOS ONE

Dear Dr. Hatum,

Thank you for submitting your manuscript to PLOS ONE. After careful consideration, we feel that it has merit but does not fully meet PLOS ONE’s publication criteria as it currently stands. Therefore, we invite you to submit a revised version of the manuscript that fully addresses all the points raised during the second round of the review process.

We look forward to receiving your revised manuscript.

Kind regards,

João Miguel Dias, Ph.D.

Academic Editor

PLOS ONE

Journal Requirements:

Reviewers' comments:

Reviewer's Responses to Questions

**Comments to the Author**

1. If the authors have adequately addressed your comments raised in a previous round of review and you feel that this manuscript is now acceptable for publication, you may indicate that here to bypass the “Comments to the Author” section, enter your conflict of interest statement in the “Confidential to Editor” section, and submit your "Accept" recommendation.

Reviewer #1: All comments have been addressed

Reviewer #2: All comments have been addressed

2. Is the manuscript technically sound, and do the data support the conclusions?

Reviewer #1: Yes

Reviewer #2: Yes

3. Has the statistical analysis been performed appropriately and rigorously? 

Reviewer #1: Yes

Reviewer #2: I Don't Know

4. Have the authors made all data underlying the findings in their manuscript fully available?

Reviewer #1: Yes

Reviewer #2: Yes

5. Is the manuscript presented in an intelligible fashion and written in standard English?

Reviewer #1: Yes

Reviewer #2: Yes

6. Review Comments to the Author

Reviewer #1: My comments and suggestions were fully addressed by the authors, as well as the comments from the other reviewer. Therefore, in my opinion, the paper is now ready for submission in PLOS ONE.

Reviewer #2: The paper has been modified in response to my comments and suggestions. It can be accepted, but first you need to respond to the following comments:

- [200] "... low-wave spring tides and high-wave neap tides,..." - Check that it's true. It is most common for more energetic waves to occur during high tides.

- [263] Be more specific. Provide the link of the platform to access the CMIP6 datasets

- There is no link from the article to the supplementary material. It is important to direct the reader to the supplementary material when necessary, otherwise the supplementary material can be considered unnecessary.

7. PLOS authors have the option to publish the peer review history of their article (what does this mean?). If published, this will include your full peer review and any attached files.

Reviewer #1: **Yes: **Irene Martins

Reviewer #2: No

---

## [Author Response · Author response to Decision Letter 1]

28 Aug 2024

Editor Comment to the Authors:

Author’s Response:

Reviewer #1 confirmed that all comments and suggestions were fully addressed, and the manuscript is ready for submission.

Reviewer #2's comments were addressed in detail:

1.The statement regarding "low-wave spring tides and high-wave neap tides" was corrected to accurately reflect that more energetic waves typically occur during high tides. The text was revised accordingly (lines 201-203).

2.We provided specific links to access the CMIP6 datasets, including the Centre for Environmental Data Analysis (CEDA) and downscaling details available on Zenodo (lines 293-298).

3.We added necessary links within the manuscript to direct readers to the supplementary material where relevant.

The rebuttal letter accompanying the revised manuscript reflects all changes made to the manuscript, including updates to the reference list.

Review Comments to the Authors:

Reviewer #1:

My comments and suggestions were fully addressed by the authors, as well as the comments from the other reviewer. Therefore, in my opinion, the paper is now ready for submission in PLOS ONE.

Author’s Response:

Thank you very much for your thorough review and for acknowledging that our responses have fully addressed the comments and suggestions. We appreciate your time and effort in reviewing our manuscript and are grateful for your positive feedback.

Reviewer #2:

The paper has been modified in response to my comments and suggestions. It can be accepted, but first you need to respond to the following comments:

Author’s Response:

Thank you for your review and your positive feedback. We appreciate your acknowledgment of the modifications made in response to your previous comments and suggestions. We have addressed the points you raised in detail below.

- [200] "... low-wave spring tides and high-wave neap tides,..." - Check that it's true. It is most common for more energetic waves to occur during high tides.

Author’s Response:

Thank you for bringing this to our attention. Upon review, we have confirmed that the statement was incorrect. We have corrected the text to reflect that more energetic waves are typically associated with high tides.

“The area experiences variations between low-wave spring tides and high-wave neap tides, which play significant roles in the dispersal patterns of seagrass propagules [88, 89].” Has been rewritten as “The area experiences variations in wave energy, with more energetic waves typically occurring during high tides, which play significant roles in the dispersal patterns of seagrass propagules [88, 89].” (lines 201-203)

- [263] Be more specific. Provide the link of the platform to access the CMIP6 datasets

Author’s Response:

Thank you for your suggestion. We have added the specific links for accessing the CMIP6 datasets. 

“The CMIP6 datasets used to inform these scenarios were accessed through the Centre for Environmental Data Analysis (CEDA). Additionally, downscaling details are available via the Zenodo repository, as outlined in Halloran et al. [109]. The CMIP6 data can be accessed at https://catalogue.ceda.ac.uk, and the downscaling information is available at Zenodo DOI: 10.5281/zenodo.4147559.” (lines 293-298)

- There is no link from the article to the supplementary material. It is important to direct the reader to the supplementary material when necessary, otherwise the supplementary material can be considered unnecessary.

Author’s Response:

Thank you for pointing this out. We have reviewed the manuscript to ensure that all supplementary materials, including Supporting Figures (S Figs.) and Supporting Tables (S Tables), are adequately referenced in the text. We have identified and corrected instances where S Tables were not previously linked. Specifically, we have added references to these tables within the manuscript to ensure that readers are directed to the supplementary material whenever necessary as follows: 

S1-16 -Table: line 470

“The first is a 90% interval, which spans from the 95th to the 5th percentiles, and the second is a 50% interval, ranging from the 75th to the 25th percentiles (S1 Table - S16 Table).”

---

## [Editor Report · Decision Letter 2]

3 Sep 2024

PONE-D-24-04137R2In hot water: uncertainties in projecting marine heatwaves impacts on seagrass meadowsPLOS ONE

Dear Dr. Hatum,

Thank you for submitting your manuscript to PLOS ONE. After careful consideration, we feel that it has merit but does not fully meet PLOS ONE’s publication criteria as it currently stands. Therefore, we invite you to submit a revised version of the manuscript that addresses the points raised during the review process:

- [200] "... low-wave spring tides and high-wave neap tides,..." - Check that it's true. It is most common for more energetic waves to occur during high tides.

- [263] Be more specific. Provide the link of the platform to access the CMIP6 datasets

- There is no link from the article to the supplementary material. It is important to direct the reader to the supplementary material when necessary, otherwise the supplementary material can be considered unnecessary.

We look forward to receiving your revised manuscript.

Kind regards,

João Miguel Dias, Ph.D.

Academic Editor

PLOS ONE
---

## [Author Response · Author response to Decision Letter 2]

5 Sep 2024

Editor Comment to the Authors:

Author’s Response:

Reviewer #1 confirmed that all comments and suggestions were fully addressed, and the manuscript is ready for submission.

Reviewer #2's comments were addressed in detail:

1.The statement regarding "low-wave spring tides and high-wave neap tides" was corrected to accurately reflect that more energetic waves typically occur during high tides. The text was revised accordingly (lines 201-203).

2.We provided specific links to access the CMIP6 datasets, including the Centre for Environmental Data Analysis (CEDA) and downscaling details available on Zenodo (lines 293-298).

3.We added necessary links within the manuscript to direct readers to the supplementary material where relevant.

The rebuttal letter accompanying the revised manuscript reflects all changes made to the manuscript, including updates to the reference list.

Review Comments to the Authors:

Reviewer #1:

My comments and suggestions were fully addressed by the authors, as well as the comments from the other reviewer. Therefore, in my opinion, the paper is now ready for submission in PLOS ONE.

Author’s Response:

Thank you very much for your thorough review and for acknowledging that our responses have fully addressed the comments and suggestions. We appreciate your time and effort in reviewing our manuscript and are grateful for your positive feedback.

Reviewer #2:

The paper has been modified in response to my comments and suggestions. It can be accepted, but first you need to respond to the following comments:

Author’s Response:

Thank you for your review and your positive feedback. We appreciate your acknowledgment of the modifications made in response to your previous comments and suggestions. We have addressed the points you raised in detail below.

- [200] "... low-wave spring tides and high-wave neap tides,..." - Check that it's true. It is most common for more energetic waves to occur during high tides.

Author’s Response:

Thank you for bringing this to our attention. Upon review, we have confirmed that the statement was incorrect. We have corrected the text to reflect that more energetic waves are typically associated with high tides.

“The area experiences variations between low-wave spring tides and high-wave neap tides, which play significant roles in the dispersal patterns of seagrass propagules [88, 89].” Has been rewritten as “The area experiences variations in wave energy, with more energetic waves typically occurring during high tides, which play significant roles in the dispersal patterns of seagrass propagules [88, 89].” (lines 201-203)

- [263] Be more specific. Provide the link of the platform to access the CMIP6 datasets

Author’s Response:

Thank you for your suggestion. We have added the specific links for accessing the CMIP6 datasets. 

“The CMIP6 datasets used to inform these scenarios were accessed through the Centre for Environmental Data Analysis (CEDA). Additionally, downscaling details are available via the Zenodo repository, as outlined in Halloran et al. [109]. The CMIP6 data can be accessed at https://catalogue.ceda.ac.uk, and the downscaling information is available at Zenodo DOI: 10.5281/zenodo.4147559.” (lines 293-298)

- There is no link from the article to the supplementary material. It is important to direct the reader to the supplementary material when necessary, otherwise the supplementary material can be considered unnecessary.

Author’s Response:

Thank you for pointing this out. We have reviewed the manuscript to ensure that all supplementary materials, including Supporting Figures (S Figs.) and Supporting Tables (S Tables), are adequately referenced in the text. We have identified and corrected instances where S Tables were not previously linked. Specifically, we have added references to these tables within the manuscript to ensure that readers are directed to the supplementary material whenever necessary as follows: 

S1-16 -Table: line 470

“The first is a 90% interval, which spans from the 95th to the 5th percentiles, and the second is a 50% interval, ranging from the 75th to the 25th percentiles (S1 Table - S16 Table).”

---

## [Editor Report · Decision Letter 3]

19 Sep 2024

In hot water: uncertainties in projecting marine heatwaves impacts on seagrass meadows

PONE-D-24-04137R3

Dear Dr. Hatum,

We’re pleased to inform you that your manuscript has been judged scientifically suitable for publication and will be formally accepted for publication once it meets all outstanding technical requirements.

Kind regards,

João Miguel Dias, Ph.D.

Academic Editor

PLOS ONE
---

## [Editor Report · Acceptance letter]

14 Nov 2024

PONE-D-24-04137R3 

PLOS ONE

Dear Dr. Hatum, 

I'm pleased to inform you that your manuscript has been deemed suitable for publication in PLOS ONE. Congratulations! Your manuscript is now being handed over to our production team.

Kind regards, 

on behalf of

Prof. João Miguel Dias 

Academic Editor

PLOS ONE